# Coevolutionary dynamics via adaptive feedback in collective-risk social dilemma game

**Linjie Liu[1,2], Xiaojie Chen[2]\*, Attila Szolnoki[3]**

[1]College of Science, Northwest A & F University, Yangling, China; [2]School of Mathematical Sciences, University of Electronic Science and Technology of China, Chengdu, China; [3]Institute of Technical Physics and Materials Science, Centre for Energy Research, Budapest, Hungary

**Abstract** Human society and natural environment form a complex giant ecosystem, where human activities not only lead to the change in environmental states, but also react to them. By using collective-risk social dilemma game, some studies have already revealed that individual contributions and the risk of future losses are inextricably linked. These works, however, often use an idealistic assumption that the risk is constant and not affected by individual behaviors. Here, we develop a coevolutionary game approach that captures the coupled dynamics of cooperation and risk. In particular, the level of contributions in a population affects the state of risk, while the risk in turn influences individuals' behavioral decision-making. Importantly, we explore two representative feedback forms describing the possible effect of strategy on risk, namely, linear and exponential feedbacks. We find that cooperation can be maintained in the population by keeping at a certain fraction or forming an evolutionary oscillation with risk, independently of the feedback type. However, such evolutionary outcome depends on the initial state. Taken together, a two-way coupling between collective actions and risk is essential to avoid the tragedy of the commons. More importantly, a critical starting portion of cooperators and risk level is what we really need for guiding the evolution toward a desired direction.

**\*For correspondence:**
xiaojiechen@uestc.edu.cn

**Competing interest:** The authors declare that no competing interests exist.

## Editor's evaluation

The paper provides a valuable, in-depth mathematical analysis of the coevolutionary dynamics resulting from a coupling of players' strategies and (collective) risk, as well as illustrative numerical simulations of the system's trajectories for different starting conditions. It is therefore a solid contribution to our understanding of how cooperation can be sustained when there is feedback between individual decisions and the global risk of disaster. This paper will be of interest to scientists working on mathematical biology/ecology, and more generally various aspects of human decision-making, the interplay between human decisions and the environment, and public goods provision.

## Introduction

Human activities constantly affect the natural environment and cause changes in its quality, which in turn affects our daily life and health conditions (*Patz et al., 2005*; *Steffen et al., 2006*; *Perc et al., 2017*; *Obradovich et al., 2018*; *Hilbe et al., 2018*; *Su et al., 2019*; *Su et al., 2022*). A well-known example is climate change, which is one of the biggest contemporary challenges of our civilization (*Parmesan and Yohe, 2003*; *Stone et al., 2013*). A large number of carbon emissions caused by human activities will exacerbate the greenhouse effect, which risks raising global temperatures to dangerous

levels. The direct consequences of global warming are the melting of glaciers and the rise in sea level, which will inevitably affect human activities (*Schuur et al., 2015*; *Obradovich and Rahwan, 2019*; *Moore et al., 2022*). Similarly, we can give more examples of coupled human and natural systems to continue this list, such as habitat destruction and the spread of infectious diseases (*Liu et al., 2001*; *Liu et al., 2007*; *Chen and Fu, 2019*; *Tanimoto, 2021*; *Chen and Fu, 2022*). At present, the importance of developing a new comprehensive framework to study the coupling between human behavior and the environment has been recognized by a number of interdisciplinary approaches (*Weitz et al., 2016*; *Chen and Szolnoki, 2018*; *Tilman et al., 2020*).

Evolutionary game theory provides a powerful theoretical framework for studying the coupled dynamics of human and natural systems (*Maynard Smith, 1982*; *Weibull, 1997*; *Stewart and Plotkin, 2014*; *Radzvilavicius et al., 2019*; *Park et al., 2020*; *Niehus et al., 2021*; *Han et al., 2021*; *Cooper et al., 2021*). Furthermore, coevolutionary game models have recognized the fact that individual payoff values are closely related to the state of the environment (*Weitz et al., 2016*; *Szolnoki and Chen, 2018*; *Chen and Szolnoki, 2018*; *Hauert et al., 2019*; *Tilman et al., 2020*; *Wang and Fu, 2020*; *Yan et al., 2021*). For example, *Weitz et al., 2016* considered the dynamical changes of the environment, which modulates the payoffs of individuals. Their results show that individual strategies and the environmental state may form a sustained cycle where strategy swing between full cooperation and full defection, while the environment state oscillates between the replete state and the depleted state. Along this line, feedback-evolving game systems with intrinsic resource dynamics (*Tilman et al., 2020*), asymmetric interactions in heterogeneous environments (*Hauert et al., 2019*), and time-delay effect (*Yan et al., 2021*) have been also investigated where periodic oscillation of strategy and environment is observed. As a general conclusion, the feedback loop between individual strategies and related environment is a key element to maintain long-term cooperation and sustainable use of resources.

Despite the mentioned efforts, the research on possible consequences of the feedback between human activity and natural systems is still in early stage. Staying at the abovementioned example, potential feedback loops between human activities and climate change exist (*Obradovich and Rahwan, 2019*). However, most scholars study these two topics, that is, human contributions to climate change and social impacts of the changing climate on human behavior, in a separated way (*Vitousek et al., 1997*; *Barfuss et al., 2020*). On the one hand, some of them usually focus on how human behaviors (use of land, oceans, fossil fuel, freshwater, etc.) affect environment (*Vitousek et al., 1997*). On the other hand, researchers who are interested in society and biology frequently focus on how environmental change will affect human behaviors (*Culler et al., 2015*; *Obradovich and Rahwan, 2019*; *Celik, 2020*). Recently, these two approaches have been merged into a single framework, called collective-risk social dilemma game, which serves as a general paradigm for studying climate change dilemmas (*Milinski et al., 2008*). Within it, a group of individuals decide whether to contribute to reach a collective goal. If the total contributions of all individuals exceed a certain threshold, then the disaster is averted and all individuals benefit from it. Otherwise, the disaster occurs with a probability (also known as the risk of collective failure), resulting in fatal economic losses for all participants. Both behavioral experiments and theoretical works show that the risk of future losses plays an important role in the evolution of cooperation (*Milinski et al., 2008*; *Santos and Pacheco, 2011*; *Chen et al., 2012a*; *Vasconcelos et al., 2013*; *Hilbe et al., 2013*; *Vasconcelos et al., 2014*; *Barfuss et al., 2020*; *Domingos et al., 2020*; *Sun et al., 2021*; *Chica et al., 2022*).

Previous studies based on the collective-risk dilemma game revealed that the risk of collective failure could affect individuals' motivation to cooperate when they face the problem of collective action, but ignored an important practical aspect. That is, human decision-making is not only affected by changes in the risk state, but also affects the level of risk (*Chen et al., 2012a*). Indeed, the risk of collective failure is lower in a highly cooperative society, but becomes significant in the opposite case. This fact is not only reflected in climate change (*Moore et al., 2022*), but also in the spread of infectious diseases (*Chen and Fu, 2022*) and vaccination (*Nichol et al., 1998*; *Chen and Fu, 2019*). Furthermore, although the risk level varies in a changing population, their relation is not necessarily straightforward. For example, a study revealed that the infection-fatality risk (IFR) of COVID-19 in India decreased linearly from June 2020 to September 2020 due to improved healthcare or increased vaccination (*Yang and Shaman, 2022*). Throughout the whole process (from March 2020 to April 2021), the statistical curve of IFR is nonlinear, that is, when the epidemic broke out, the value of IFR

remained at a high level, and then with the increase in vaccination or the improvement in healthcare, the IFR value gradually decreased, then flattened and remained at a low level (*Yang and Shaman, 2022*). On the other hand, the change in risk is bound to affect individuals' decision-making, which has been confirmed in behavioral experiments and theoretical research (*Milinski et al., 2008*; *Pacheco et al., 2014*). Though potential feedback loops between strategy and risk of future losses are already recognized, a study focusing on their direct interaction is missing. Furthermore, it is still an open question whether the character of feedback mechanism plays an essential role in the final evolutionary outcome. Hence, how the impacts of risk on human systems might, in turn, alter the future trajectories of human decision-making remains largely unexplored.

To fill this gap, we propose a coupled coevolutionary game framework based on the collective-risk dilemma to describe reciprocal interactions and feedbacks between decision-making procedure of individuals and risk. In particular, we assume that the increasing free-riding behaviors will slowly increase the risk of collective failure, and the resulting high-risk level will in turn stimulate individual contributions. However, the increase in contribution will gradually reduce the risk of collective failure, and the resulting low-risk level will promote the prevalence of free-riding behaviors again. This general feedback loop is illustrated in *Figure 1*. Importantly, we respectively consider two conceptually different feedback protocols describing the effect of strategy on risk. Namely, both linear and highly nonlinear (exponential) feedback forms are checked. Our analysis identifies the conditions for the existence of stable interior equilibrium and stable limit cycle dynamics in both cases.

## Materials and methods

### Collective-risk social dilemma game

We consider an infinite well-mixed population in which $N$ individuals are selected randomly to form a group for playing the collective-risk social dilemma game. Each individual in the group has an initial endowment $b$ and can choose one of the two strategies, that is, cooperation and defection. Cooperators will contribute an amount $c$ to the common pool, whereas defectors contribute nothing. The remaining endowments of all individuals can be preserved if the overall number of cooperators exceeds a threshold value $M$, where $1 < M < N$ (*Milinski et al., 2008*; *Santos and Pacheco, 2011*). Otherwise, individuals will lose all their endowments with a probability $r$, which characterizes the risk level of collective failure. Accordingly, the payoffs of cooperators and defectors in a group of size $N$ with $j_C$ cooperators and $N - j_C$ defectors can be summarized as

$$P_C = b\theta(j_C + 1 - M) + (1 - r)b[1 - \theta(j_C + 1 - M)] - c, \tag{1}$$

$$P_D = b\theta(j_C - M) + (1 - r)b(1 - \theta(j_C - M)), \tag{2}$$

where $\theta(x)$ is the Heaviside function, that is, $\theta(x) = 0$ if $x < 0$, being one otherwise. Here, we would like to note that the collective-risk social dilemma is a kind of public goods games, which are a special and extended version of Donor & Recipient game by referring to the concept of universal dilemma strength (*Wang et al., 2015*; *Ito and Tanimoto, 2018*; *Tanimoto, 2021*). However, following previous work on collective-risk social dilemma (*Santos and Pacheco, 2011*), we retain the parameters mentioned above for the sake of convenience, instead of replacing them with the universal dilemma strength.

To analyze the evolutionary dynamics of strategies in an infinite population, we use replicator equations to describe the time evolution of cooperation (*Taylor and Jonker, 1978*; *Schuster and Sigmund, 1983*). Accordingly, we have

$$\dot{x} = x(1 - x)(f_C - f_D),$$

where $x$ denotes the frequency of cooperators in the population, while $f_C$ and $f_D$ respectively denote the average payoffs of cooperators and defectors, which can be calculated as

$$f_C = \sum_{jc=0}^{N-1} \binom{N-1}{jc} x^{jc}(1 - x)^{N-jc} P_C,$$
$$f_D = \sum_{jc=0}^{N-1} \binom{N-1}{jc} x^{jc}(1 - x)^{N-jc} P_D,$$

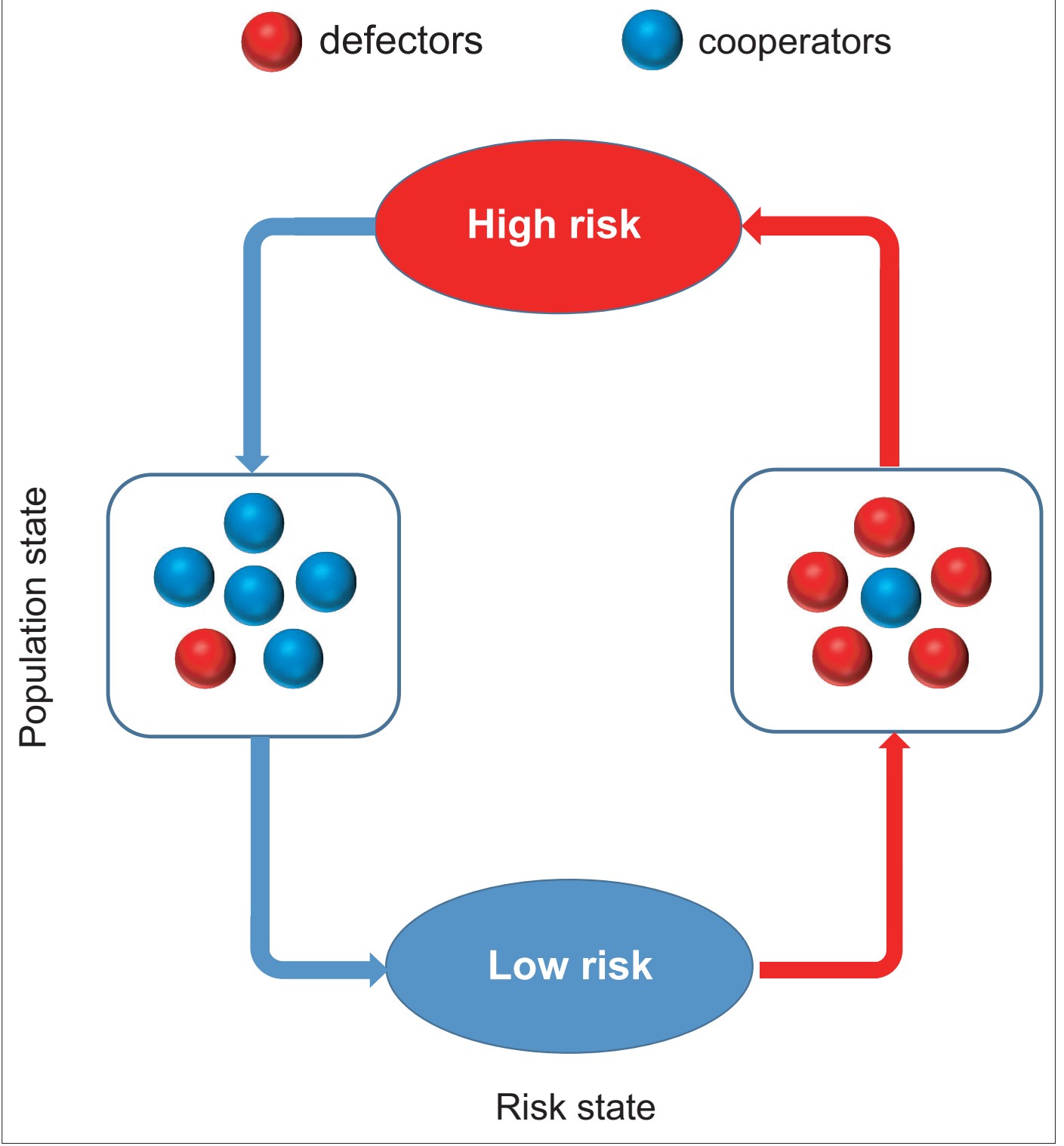

**Figure 1.** Coevolutionary feedback loop of population and risk states in the coupled game system. The meaning of colors is explained in the legend on the top.

where $P_C$ and $P_D$ are shown in *Equations 1 and 2*. After some calculations, the difference between the average payoffs of cooperators and defectors can be written as

$$f_C - f_D = \binom{N-1}{M-1}x^{M-1}(1-x)^{N-M}rb - c.$$

In the above replicator equation, we describe a game-theoretic interaction involving the risk of collective failure, which is a positive constant in previous works (*Santos and Pacheco, 2011*; *Chen et al., 2012a*). Here, we are focusing on a dynamical system where there is feedback between strategic behaviors and risk. In particular, the impact of strategies on the risk level is channeled through a function $U(x, r)$, which depends on both key variables. Then by using the general form of the feedback, the coevolutionary dynamics can be written as

$$\begin{cases} \varepsilon\dot{x} = x(1-x)[\binom{N-1}{M-1}x^{M-1}(1-x)^{N-M}rb - c], \\ \dot{r} = U(x, r), \end{cases} \quad (3)$$

where $\varepsilon$ denotes the relative speed of strategy update dynamics (*Weitz et al., 2016*), such that when $0 < \varepsilon \ll 1$ the strategies evolve significantly faster than the change in the risk level. In the following, we consider both linear and nonlinear forms of feedback describing the effect of strategy distribution on the evolution of risk.

## Linear effect of strategy on risk

In the first case, we assume that the effect of strategies on the risk level takes a linear form, which is the most common form that can be used to describe the characteristic attributes between key variables. Just to illustrate it by a specific example, the probability of influenza infection among individuals who have not been vaccinated decreases linearly with the increase in vaccine coverage (*Vardavas et al., 2007*). Here, we consider that the value of risk decreases linearly with the increase in cooperation level. Furthermore, by following the work of *Weitz et al., 2016*, we can write the dynamical equation of risk as

$$\dot{r} = r(1-r)[u(1-x) - x], \quad (4)$$

where $u(1-x) - x$ denotes the increase in risk by the defection level at rate $u$ and the decrease by the fraction of cooperators at relative rate one. Then the dynamical system is described by the following equation:

$$\begin{cases} \varepsilon\dot{x} = x(1-x)[\binom{N-1}{M-1}x^{M-1}(1-x)^{N-M}rb - c] \\ \dot{r} = r(1-r)[u(1-x) - x]. \end{cases} \quad (5)$$

## Exponential effect of strategy on risk

To complete our study, we also apply a nonlinear form of feedback function. The most plausible choice is when the risk level depends exponentially on the population state. To be more specific, we consider that the risk will decrease when the frequency of cooperators in the population exceeds a certain threshold value $T$. Otherwise, the risk level will increase. Such a scenario is suitable for describing climate change and the spread of infectious diseases, in which the risk can increase sharply, such as the occurrence of extreme weather (*Eckstein et al., 2021*) or a sudden outbreak of an epidemic in a region (*Yang and Shaman, 2022*). Here, we use the sigmoid function to describe the effect of strategy on the risk state (*Boza and Számadó, 2010*; *Chen et al., 2012b*; *Couto et al., 2020*), which can be written as

$$\dot{r} = r(1-r)[\frac{1}{1+e^{\beta(x-T)}} - \frac{1}{1+e^{-\beta(x-T)}}], \quad (6)$$

where $\beta$ characterizes the steepness of the function and $r(1-r)$ ensures that the risk state remains in the $[0, 1]$ domain. For convenience, we introduce the variable $\xi = x - T$ and the function $B(\xi) = \frac{1}{1+e^{\beta\xi}} - \frac{1}{1+e^{-\beta\xi}}$. Thus we have $\dot{r} = r(1-r)B(\xi)$. When $\beta = 0$, we know that $B(\xi) = 0$. In this

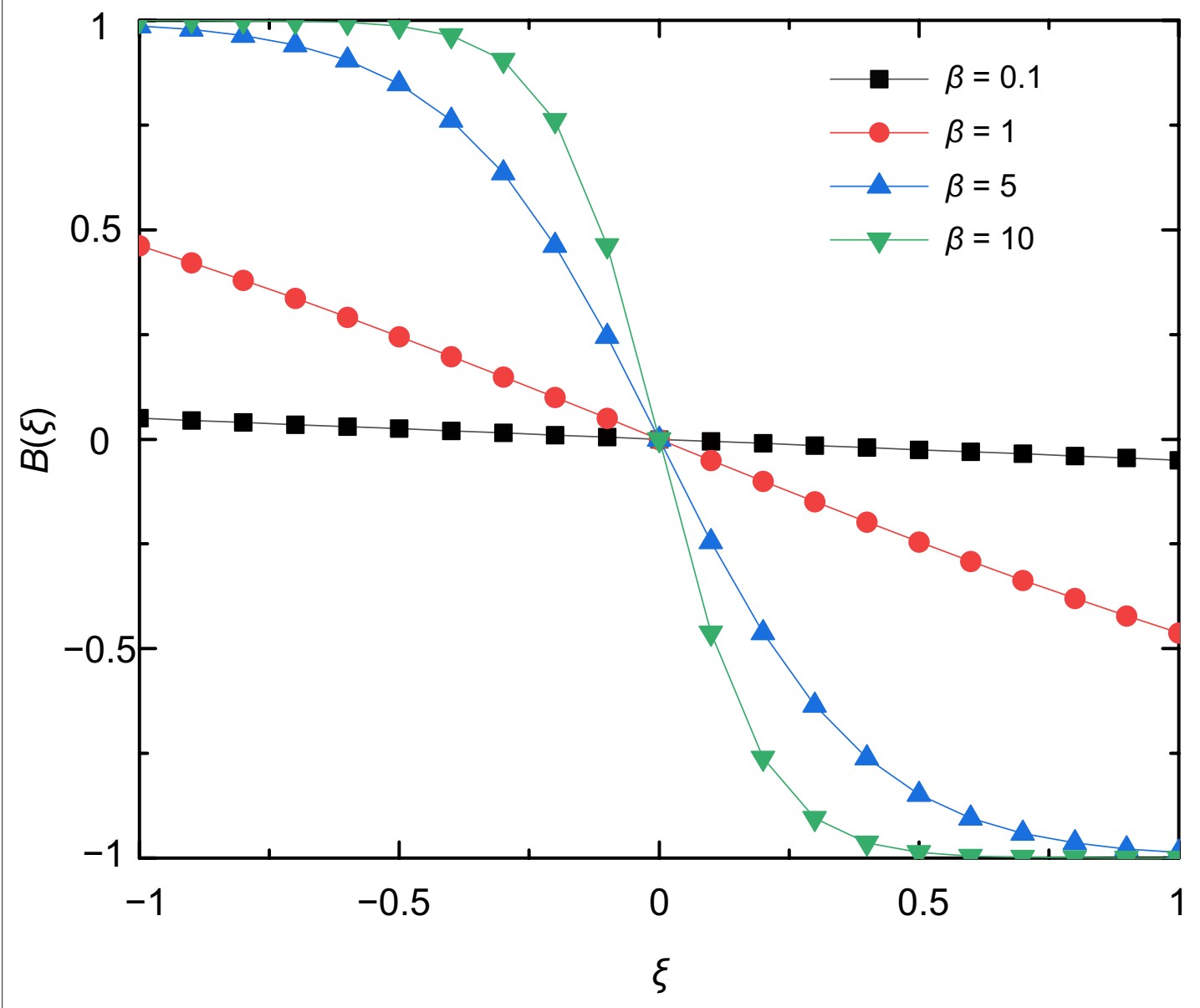

**Figure 2.** Feedback equation $B(\xi)$ varies with $\xi$ for different values of $\beta$. The parameter $\beta$ determines the steepness of the curves. When the value of $\beta$ is small, the $B(\xi)$ function is almost constant or decays linearly by increasing $\xi$. For larger $\beta$ values, the shape of $B(\xi)$ approaches a step-like form. In this parameter area, the risk level depends sensitively on whether the group cooperation exceeds the threshold $T$ value or not.

situation, strategies have no effect on the risk level. For $\beta = +\infty$, the function $B(\xi)$ becomes steplike so that the risk will decrease only if the frequency of cooperators in the group exceeds the threshold $T$. Otherwise, the risk level remains high. To study the consequence of a proper feedback effect, we apply a finite $\beta > 0$ value. In *Figure 2*, we illustrate how $B(\xi)$ varies with $\xi$ for four different values of $\beta$.

Accordingly, the feedback-evolving dynamical system where the effect of strategies on the risk state is expressed by the exponential form can be written as

$$\begin{cases} \varepsilon \dot{x} = x(1-x)[\binom{N-1}{M-1} x^{M-1}(1-x)^{N-M} rb - c] \\ \dot{r} = r(1-r)[\dfrac{1}{1+e^{\beta(x-T)}} - \dfrac{1}{1+e^{-\beta(x-T)}}]. \end{cases} \tag{7}$$

**Table 1.** Notation symbols and meanings in our work.

| Symbol | Meaning |
| --- | --- |
| $N$ | Group size |
| $b$ | Initial endowment |
| $c$ | Cost of cooperation |
| $r$ | Risk |
| $M$ | Collective goal |
| $\varepsilon$ | Feedback speed |
| $u$ | Growth rate of risk with the proportion of defectors |
| $T$ | Threshold value of cooperation |
| $\beta$ | Steepness parameter |
| $x$ | Frequency of cooperation |

Here, in order to help readers to overview easily all the parameters and variables introduced in our work, we present them in **Table 1**. In the following section, we respectively investigate the coevolutionary dynamics of strategy and risk when considering linear and exponential feedback forms. We note that the details of theoretical analysis can be found in Appendix 1.

## Results

### System I: Coevolutionary dynamics with linear feedback

We first consider the case of linear feedback. More precisely, we assume that the risk value of collective failure will decrease linearly with the increase in cooperation and increase linearly with the increase in defection level. The resulting dynamical system is presented in **Equation 5**.

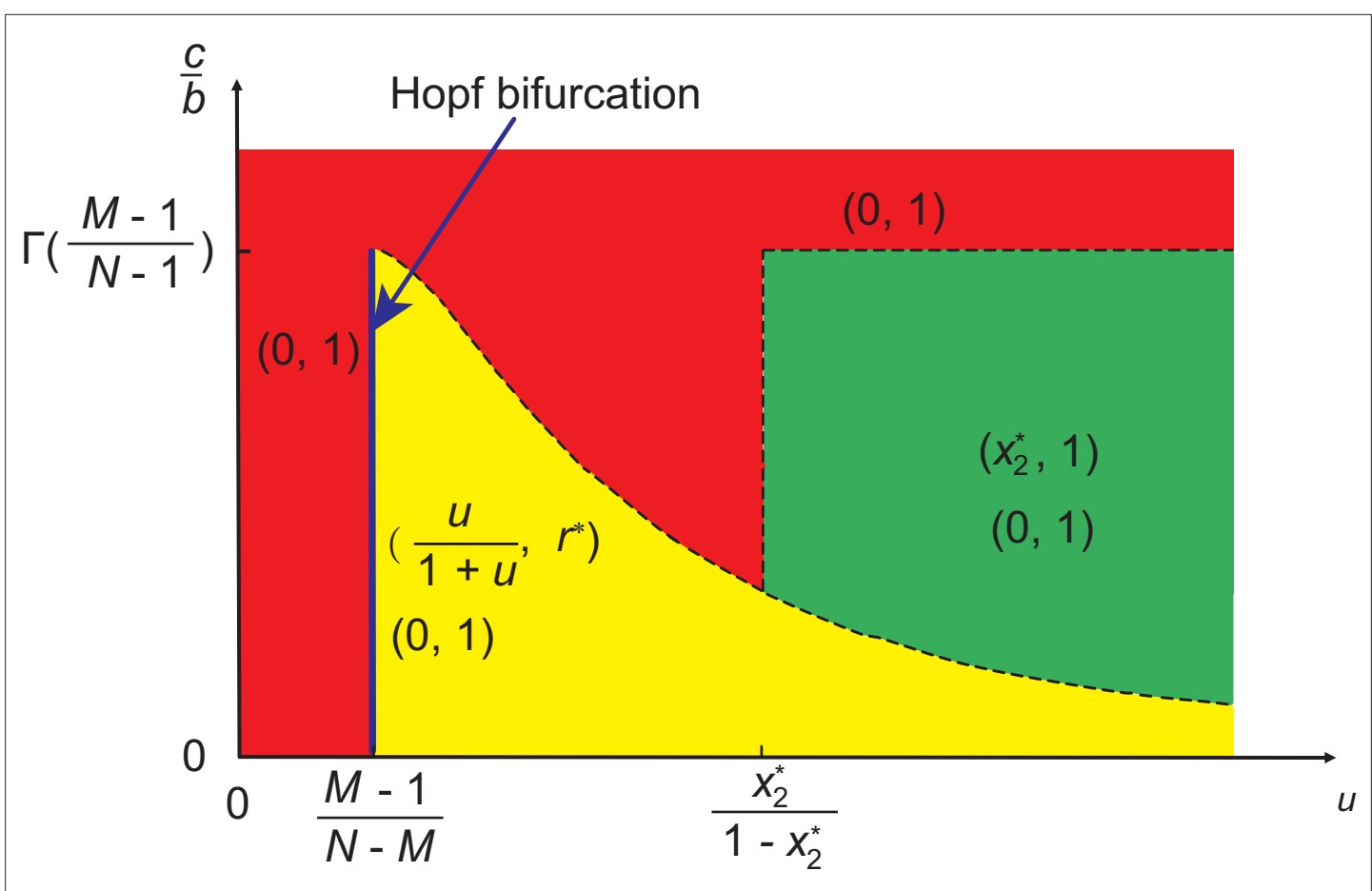

**Figure 3.** Representative plot of stable evolutionary outcomes in System I when linear strategy feedback on risk level is assumed. Different colors are used to distinguish the stability of different equilibrium points in the parameter space $(u, \frac{c}{b})$. The blue line indicates that the system undergoes a Hopf bifurcation at $u = \frac{M-1}{N-M}$. Here, $x_2^*$ is the real root of the equation $\Gamma(x) = \frac{c}{b}$, where $\Gamma(x) = \binom{N-1}{M-1}x^{M-1}(1-x)^{N-M}$, and $(\frac{u}{1+u}, r^*)$ is the interior fixed point where $r^* = \frac{c}{\binom{N-1}{M-1}(\frac{u}{1+u})^{M-1}(\frac{1}{1+u})^{N-M}b}$. The dashed curve represents that the value of $\Gamma(\frac{u}{1+u})$ changes with $u$ when $u > \frac{M-1}{N-M}$. The horizontal dashed line represents that $\Gamma(\frac{M-1}{N-1}) = \frac{c}{b}$ when $u > \frac{x_2^*}{1-x_2^*}$. The vertical dashed line represents that $u = \frac{x_2^*}{1-x_2^*}$ when $\Gamma(x_2^*) < \frac{c}{b} < \Gamma(\frac{M-1}{N-1})$.

After some calculations, we find that this equation system has at most seven fixed points, which are $(0,0), (0,1), (1,0), (1,1), (\frac{u}{1+u}, r^*), (x_1^*, 1)$, and $(x_2^*, 1)$, where $r^* = \frac{c}{\binom{N-1}{M-1}(\frac{u}{1+u})^{M-1}(\frac{1}{1+u})^{N-M}b}$, $x_1^*$ and $x_2^*$ are the real roots of the equation $\binom{N-1}{M-1}x^{M-1}(1-x)^{N-M}b = c$. We further perform theoretical analysis for these equilibrium points, as provided in Appendix 1. In order to describe the stable states of System I for the complete parameter regions, we present a schematic plot in the parameter space $(u, \frac{c}{b})$, as shown in *Figure 3*. We use different colors to distinguish the evolutionary outcomes for specific pairs of key parameters. In the following, we discuss the representative results in detail.

## System I has an interior equilibrium point

When $\binom{N-1}{M-1}(\frac{u}{1+u})^{M-1}(\frac{1}{1+u})^{N-M}b > c$, we know that our coevolutionary system has an interior fixed point. According to its stability, we can distinguish three subcases here. Namely, when $u > \frac{M-1}{N-M}$, then the existing interior fixed point is stable. Besides, since $\binom{N-1}{M-1}(\frac{M-1}{N-1})^{M-1}(1-\frac{M-1}{N-1})^{N-M}b > c$, there exist seven fixed points in the system, namely, $(0,0), (0,1), (1,0), (1,1), (\frac{u}{1+u}, r^*), (x_1^*, 1)$, and $(x_2^*, 1)$. Here, only $(0,1)$ and $(\frac{u}{1+u}, r^*)$ are stable (marked by the yellow area in *Figure 3*). Besides, we provide numerical examples to illustrate the above theoretical analysis (see the top row of *Figure 4*). We find that bistable dynamics can appear, that is, depending on the initial conditions the system will evolve to one of two stable equilibria: here, $(0,1)$, which is the undesirable full defection equilibrium, or the interior fixed point suggests that cooperation can be maintained at a high level when the value of risk exceeds an intermediate value. Furthermore, we note that the results are not affected qualitatively by the feedback speed in any of the cases (see *Appendix 1—figure 1* in Appendix 1).

If the enhancement rate of risk caused by defection drops to a certain threshold, namely, $u = \frac{M-1}{N-M}$, a Hopf bifurcation takes place, which is supercritical (marked by the blue line in *Figure 3*). In this situation, System I has all seven fixed points. As analyzed in Appendix 1, only $(0,1)$ is stable. Furthermore, we provide numerical examples to illustrate our theoretical analysis (see the second row of *Figure 4*). We find that the system is bistable: depending on the initial fractions of cooperators and risk, the system can evolve either to a high-risk state without cooperation or to a limit cycle where the frequencies of cooperation and risk show periodic oscillations.

When the enhancement rate of risk caused by defection is weak and meets $u < \frac{M-1}{N-M}$ condition, then the interior fixed point is unstable. Besides, since $\binom{N-1}{M-1}(\frac{M-1}{N-1})^{M-1}(1-\frac{M-1}{N-1})^{N-M}b > c$, there exist all seven fixed points. According to the theoretical analysis presented in Appendix 1, only $(0,1)$ fixed point is stable. In the third row of *Figure 4*, we present some representative numerical examples. They show that all trajectories in the state space terminate at the fixed point $(0,1)$, which is consistent with our theoretical results. This means that no individual chooses to contribute to the common pool, leading to the failure of collective action, and finally, all individuals inevitably lose all their endowments.

## System I has no interior equilibrium point

The alternative case is when there is no interior fixed point, namely, $\binom{N-1}{M-1}(\frac{u}{1+u})^{M-1}(\frac{1}{1+u})^{N-M}b \leq c$. In this situation, when $\binom{N-1}{M-1}(\frac{M-1}{N-1})^{M-1}(1-\frac{M-1}{N-1})^{N-M}b > c$, System I has six fixed points, which are $(0,0), (0,1), (1,0), (1,1), (x_1^*, 1)$, and $(x_2^*, 1)$, respectively. The theoretical analysis, presented in Appendix 1, shows that $(0,0), (1,0), (1,1)$, and $(x_1^*, 1)$ are unstable, $(0,1)$ is stable, and $(x_2^*, 1)$ is stable for $x_2^* < \frac{u}{1+u}$ (shown by the green area in *Figure 3*). In the bottom row of *Figure 4*, we provide some numerical examples to illustrate our theoretical results. The phase plane dynamics show that most trajectories in phase space converge to the stable equilibrium point $(x_2^*, 1)$, which suggests that driven by the high risk of future loss, most individuals will contribute to the common pool. Besides, the remaining trajectories in the phase space will converge to the fixed point $(0,1)$, which means a complete failure when all individuals lose all remaining endowments.

Furthermore, we prove that the fixed point $(x_2^*, 1)$ is unstable when $x_2^* > \frac{u}{1+u}$ in Appendix 1. For the special case of $x_2^* = \frac{u}{1+u}$, we find that one eigenvalue of the Jacobian matrix at $(x_2^*, 1)$ is zero and the other one is negative. We provide the stability analysis of this fixed point by using the center manifold theorem (*Khalil, 1996*). When $\binom{N-1}{M-1}(\frac{M-1}{N-1})^{M-1}(1-\frac{M-1}{N-1})^{N-M}b \leq c$, $(0,1)$ is the only stable equilibrium point of the System I.

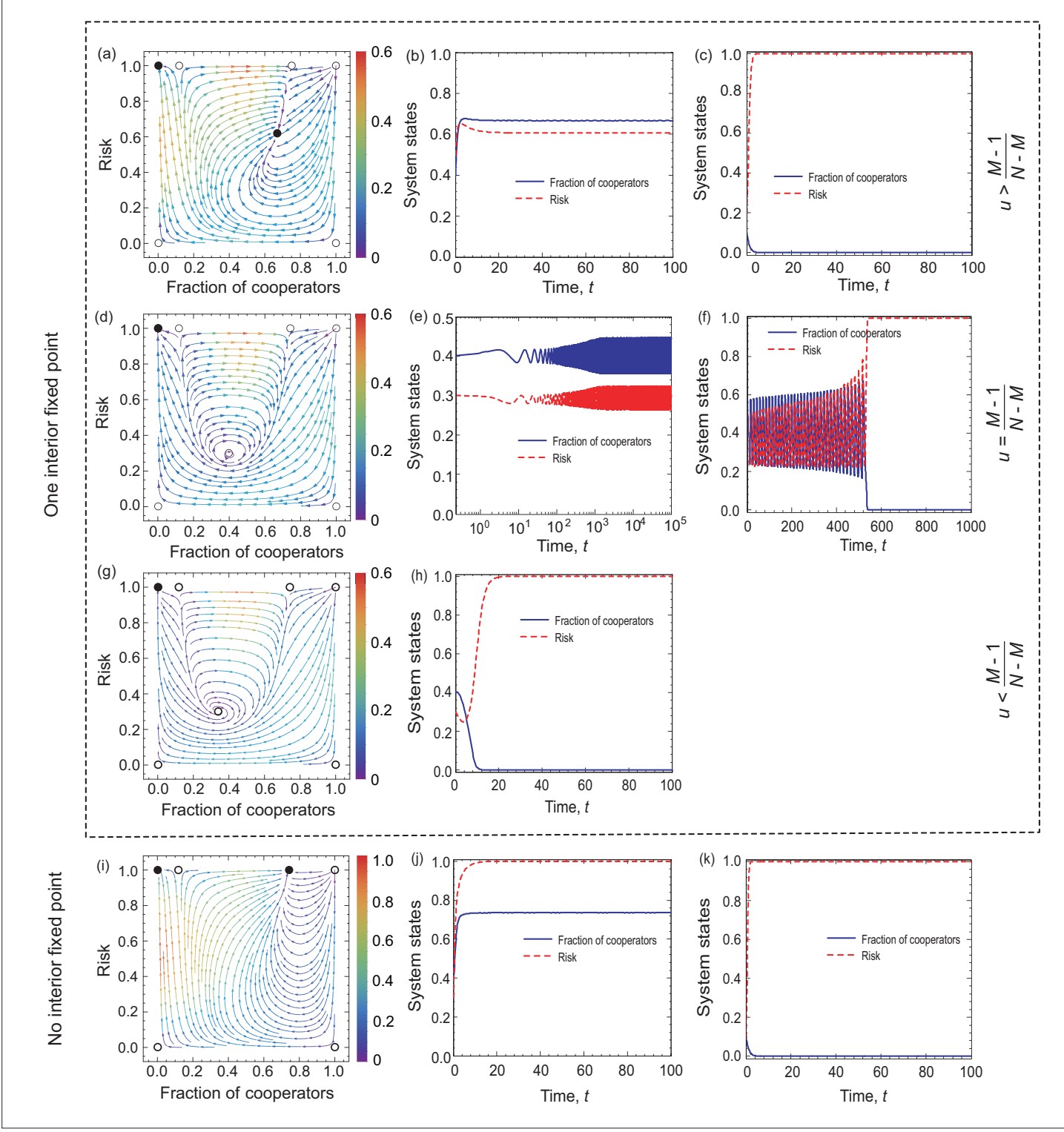

**Figure 4.** Coevolutionary dynamics on phase planes and temporal dynamics of System I when linear feedback is considered. Filled circles represent stable and open circles denote unstable fixed points. The arrows provide the most likely direction of evolution and the continuous color code depicts the speed of convergence in which red denotes the highest speed, while purple represents the lowest speed of transition. On the right-hand side, blue solid line and red dash line respectively denote the fraction of cooperation and the risk level, as indicated in the legend. The first three rows show the coevolutionary dynamics when $u > \frac{M-1}{N-M}$, $u = \frac{M-1}{N-M}$, and $u < \frac{M-1}{N-M}$, respectively. The bottom row shows coevolutionary dynamics when $\binom{N-1}{M-1}(\frac{u}{1+u})^{M-1}(\frac{1}{1+u})^{N-M}b < c$. Parameters are $N = 6, c = 0.1, b = 1, u = 2, \varepsilon = 0.1, M = 3$ in panel (**a**). The initial conditions are $(x, r) = (0.4, 0.3)$ in

*Figure 4 continued on next page*

*Figure 4 continued*

panel (**b**) and $(x, r) = (0.1, 0.1)$ in panel (**c**). $N = 6, c = 0.1, b = 1, u = \frac{2}{3}, \varepsilon = 0.1, M = 3$ in panel (**d**). The initial conditions are $(x, r) = (0.4, 0.3)$ in panel (**e**) and $(x, r) = (0.4, 0.5)$ in panel (**f**). $N = 6, c = 0.1, b = 1, u = 0.5, \varepsilon = 0.1, M = 3$ in panel (**g**). The initial conditions are $(x, r) = (0.4, 0.3)$ in panel (**h**). $N = 6, c = 0.1, b = 1, u = 4, \varepsilon = 0.1, M = 3$ in panel (**i**). The initial conditions are $(x, r) = (0.4, 0.3)$ in panel (**j**) and $(x, r) = (0.1, 0.1)$ in panel (**k**).

## System II: Coevolutionary dynamics with exponential feedback

In this section, we consider the case of exponential feedback. Here, there are at most seven equilibrium points of the replicator *Equation 7*. Namely, $(x, y) = (0, 0),\ (0, 1),\ (1, 0),\ (1, 1),\ (x_1^*, 1),\ (x_2^*, 1),$ and $(T, \frac{c}{\binom{N-1}{M-1} T^{M-1}(1-T)^{N-M} b})$, in which $x_1^*$ and $x_2^*$ satisfy the equation $\binom{N-1}{M-1} x^{M-1} (1 - x)^{N-M} b = c$ and $x_1^* < \frac{M-1}{N-1} < x_2^*$ (*Santos and Pacheco, 2011*). For convenience, we set $\bar{r} = \frac{c}{\binom{N-1}{M-1} T^{M-1}(1-T)^{N-M} b}$. Here, the first six equilibria are boundary fixed points, and the last one is an interior fixed point. In Appendix

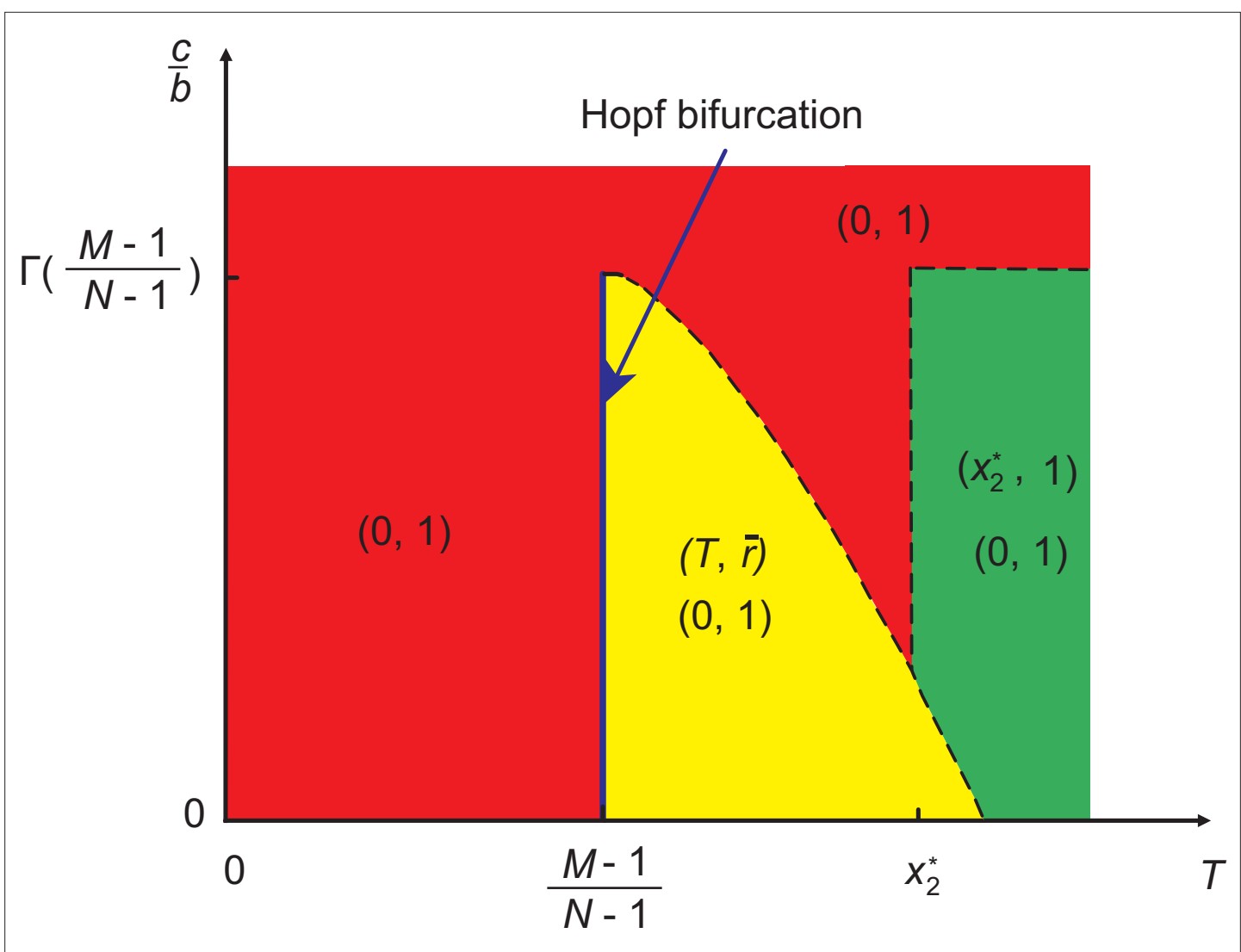

**Figure 5.** A representative diagram about stable solutions of System II when strategy feedback on risk level is exponential. We use different colors to distinguish the stability of equilibrium points in the parameter space $(T, \frac{c}{b})$. The blue line indicates that the system undergoes a Hopf bifurcation at $T = \frac{M-1}{N-1}$. Here, $(T, \bar{r})$ is the interior fixed point where $\bar{r} = \frac{c}{\binom{N-1}{M-1} T^{M-1}(1-T)^{N-M} b}$. The dashed curve represents that the value of $\Gamma(T)$ changes with $T$ when $T > \frac{M-1}{N-1}$. The horizontal dashed line represents that $\Gamma(\frac{M-1}{N-1}) = \frac{c}{b}$ when $T > x_2^*$. The vertical dashed line represents that $T = x_2^*$ when $\Gamma(x_2^*) < \frac{c}{b} < \Gamma(\frac{M-1}{N-1})$.

2, we analyze the stability of these equilibria under four different parameter ranges by evaluating the sign of the eigenvalues of the Jacobian (**Khalil, 1996**). The basins of each solution in parameter space $(T, \frac{c}{b})$ are shown in **Figure 5**. In the following, we will discuss the evolutionary outcomes depending on whether System II has an interior equilibrium point.

## System II has an interior equilibrium point

In this case $c < \binom{N-1}{M-1}T^{M-1}(1-T)^{N-M}b$, there are three typical dynamical behaviors for the evolution of cooperation and risk according to the stability conditions of the interior equilibrium point (for details, see Appendix 2).

When $T > \frac{M-1}{N-1}$, the interior fixed point is stable. Besides, since $\binom{N-1}{M-1}(\frac{M-1}{N-1})^{M-1}(1-\frac{M-1}{N-1})^{N-M}b - c > 0$, there exist two boundary fixed points, which are $(x_1^*, 1)$ and $(x_2^*, 1)$. Thus the system has seven fixed points, which are $(0,0), (0,1), (1,0), (1,1), (x_1^*, 1), (x_2^*, 1)$, and $(T, \bar{r})$. From the Jacobian matrices, we can conclude that the fixed points $(0,0), (0,1), (1,0), (1,1), (x_1^*, 1)$, and $(x_2^*, 1)$ are unstable, while $(0,1)$ and $(T, \bar{r})$ are stable. The latter case is shown in the top row of **Figure 6**, where we plot the phase plane and temporal dynamics of the system. It suggests that there is a stable interior fixed point, and most trajectories in phase space converge to this nontrivial solution, which means that the system can evolve into a state where the risk is kept at a low level and almost half of the individuals contribute to the common pool. The remaining trajectories in the phase space will converge to the alternative destination in which the risk level becomes particularly high and cooperators disappear.

When $T = \frac{M-1}{N-1}$, the eigenvalues of Jacobian matrix at the interior fixed point are a purely imaginary conjugate pair. Then, according to the Hopf bifurcation theorem (**Kuznetsov, 1998**; **Guckenheimer and Holmes, 2013**), the system undergoes a Hopf bifurcation at $T = \frac{M-1}{N-1}$ and a limit cycle encircling around interior equilibrium emerges. By calculating the first Lyapunov coefficient, we can evaluate that the limit cycle is stable (see Appendix 2). Besides, there exist two boundary fixed points, $(x_1^*, 1)$ and $(x_2^*, 1)$, because $\binom{N-1}{M-1}(\frac{M-1}{N-1})^{M-1}(1-\frac{M-1}{N-1})^{N-M}b - c > 0$. Thus the system has all seven fixed points. As we discuss in Appendix 2, only the fixed point $(0,1)$ is stable. A representative numerical example is shown in the second row of **Figure 6**, which is conceptually similar to those we observed for System I. More precisely, the population either converges toward a limit cycle in the interior space or arrives to the undesired $(0,1)$ point where there are no cooperators, but just high risk.

The interior fixed point is unstable when $T < \frac{M-1}{N-1}$. Besides, there are two boundary fixed points, $(x_1^*, 1)$ and $(x_2^*, 1)$, because $\binom{N-1}{M-1}(\frac{M-1}{N-1})^{M-1}(1-\frac{M-1}{N-1})^{N-M}b - c > 0$. In this situation, the system has all seven fixed points. Theoretical analysis, presented in Appendix 2, confirms that only $(0,1)$ is stable. This is illustrated in the third row of **Figure 6** where all trajectories terminate in the mentioned point, signaling that the tragedy of the commons state is inevitable.

## System II has no interior equilibrium point

When $c \geq \binom{N-1}{M-1}T^{M-1}(1-T)^{N-M}b$, there is no interior fixed point in System II. In this case, when $\binom{N-1}{M-1}(\frac{M-1}{N-1})^{M-1}(1-\frac{M-1}{N-1})^{N-M}b - c < 0$, there are four equilibrium points, namely, $(0,0), (0,1), (1,0), (1,1)$, where $(0,1)$ is stable. When $\binom{N-1}{M-1}(\frac{M-1}{N-1})^{M-1}(1-\frac{M-1}{N-1})^{N-M}b - c > 0$, there exist two boundary fixed points, $(x_1^*, 1)$ and $(x_2^*, 1)$. Altogether, the system has six fixed points, which are $(0,0), (0,1), (1,0), (1,1), (x_1^*, 1)$, and $(x_2^*, 1)$. As we discuss in Appendix 2, the fixed points $(0,0), (1,0), (1,1), (x_1^*, 1)$ are unstable, while $(0,1)$ is stable. In the special case of $x_2^* < T$, the fixed point $(x_2^*, 1)$ becomes stable, which suggests that there is a significant cooperation at a high risk level. A representative numerical illustration is shown in the bottom row of **Figure 6**, signaling the importance of the initial conditions because the trajectories converge either to the fixed point $(0,1)$ or to $(x_2^*, 1)$.

## Discussion

Human behavior and the natural environment are inextricably linked. Motivated by this fact, rapidly growing research efforts have recognized the importance of developing a new comprehensive framework to study the coupled human–environment ecosystem (**Stern, 1993**; **Liu et al., 2007**; **Farahbakhsh et al., 2022**). Starting from the powerful concept of coevolutionary game theory, several works focus on depicting the reciprocal interactions and feedback between human behaviors and natural environment – both the impact of human behaviors on nature and the effects of environment on human behaviors (**Weitz et al., 2016**; **Chen and Szolnoki, 2018**; **Tilman et al., 2020**). Along this

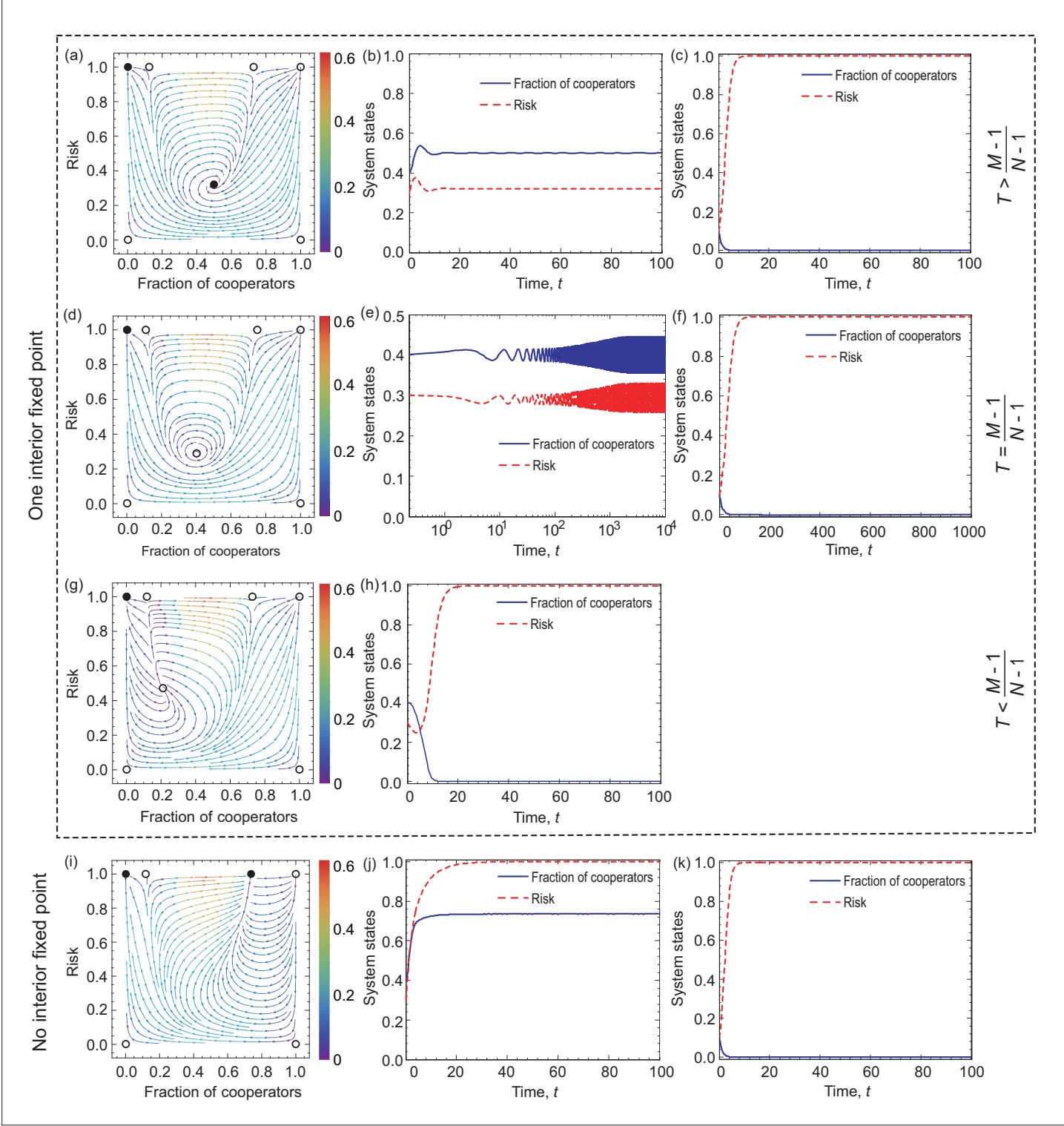

**Figure 6.** Coevolutionary dynamics on phase planes and temporal dynamics of System II when exponential feedback is assumed. Filled circles represent stable and open circles denote unstable fixed points. The arrows provide the most likely direction of evolution and the continuous color code depicts the speed of convergence in which red denotes the highest speed, while purple represents the lowest speed of transition. Blue solid line and red dash line respectively denote the fraction of cooperation and the risk level, as indicated in the legend. The first three rows show the coevolutionary dynamics when $T > \frac{M-1}{N-1}$, $T = \frac{M-1}{N-1}$, and $T < \frac{M-1}{N-1}$, respectively. The bottom row shows the case when $c > \binom{N-1}{M-1} T^{M-1}(1-T)^{N-M}b$. Parameters are $N = 6, c = 0.1, b = 1, T = 0.5, \varepsilon = 0.1, M = 3$ in panel (**a**). The initial conditions are $(x, r) = (0.4, 0.3)$ in panel (**b**) and $(x, r) = (0.1, 0.1)$ in panel (**c**). $N = 6, c = 0.1, b = 1, T = 0.4, \varepsilon = 0.1, M = 3$ in panel (**d**). The initial conditions are $(x, r) = (0.4, 0.3)$ in panel (**e**) and $(x, r) = (0.4, 0.5)$ in panel

*Figure 6 continued on next page*

*Figure 6 continued*

(**f**). $N = 6, c = 0.1, b = 1, T = 0.2, \varepsilon = 0.1, M = 3$ in panel (**g**). The initial conditions are $(x, r) = (0.4, 0.3)$ in panel (**h**). $N = 6, c = 0.1, b = 1, T = 0.8,$ $\varepsilon = 0.1, M = 3$ in panel (**i**). The initial conditions are $(x, r) = (0.4, 0.3)$ in panel (**j**) and $(x, r) = (0.1, 0.1)$ in panel (**k**).

research line, we have developed a feedback-evolving game framework to study the coevolutionary dynamics of strategies and environment based on collective-risk dilemmas. Here, the environmental state is no longer a symbol of resource abundance, but depicts the risk level of collective failure. More precisely, we assume that the frequencies of strategies directly affect the risk level and reversely, the change in risk state stimulates individual behavioral decision-making. Importantly, we have explored both linear and highly nonlinear feedback mechanisms that characterize the link between the main system variables.

In particular, we have incorporated the strategies-risk feedback mechanism into replicator dynamics and explored the possible consequences of coevolutionary dynamics. We have shown that sustainable cooperation level can be reached in the population in two different ways. First, the coevolutionary dynamics can converge to a fixed point. This fixed point can be in the interior, indicating that the frequency of cooperators and the level of risk can be respectively stabilized at a certain level, or at the boundary, indicating that high-level cooperation can be maintained even at a significantly high-risk environment. Second, the system has a stable limit cycle where persistent oscillations in strategy and risk state can appear. In addition, we have found that the above-described evolutionary outcomes do not depend significantly on the character of feedback mechanism of how strategy change affects on risk level. No matter it is linear or nonlinear, what really counts is the existence of the proper feedback. Importantly, we have theoretically identified those conditions that are responsible for the final dynamical outcomes.

In this work, we introduce a two-way coupling between strategy and environment. Indeed, the effect of a two-way interplay between environments and strategy has been involved in previous works. For example, Hilbe et al. considered that the public resource is changeable and depends on strategic choices of individuals (*Hilbe et al., 2018*). By analyzing the stochastic dynamics of the system, they found that the interplay between reciprocity and payoff feedback can be crucial for cooperation. And they considered this two-way interplay between environments and strategy in repeated stochastic game with discrete time steps. Differently, in our work we focus on one-shot collective-risk social dilemma with such two-way interplay. We find that a two-way coupling between collective actions and risk is essential to avoid the tragedy of the commons.

Previous theoretical studies have revealed that the coevolutionary game models describing the complex interactions between collective actions and environment can produce periodic oscillation dynamics (*Weitz et al., 2016*; *Tilman et al., 2020*). Although our feedback-evolving game model can also produce persistent oscillations, there are some differences. In particular, we have theoretically proved that Hopf bifurcation can take place and a stable limit cycle can appear in the system, which is different from the heteroclinic cycle dynamics reported by *Weitz et al., 2016*. Besides, we have found that the existence of a limit cycle does not depend on the speed of coupling (see *Appendix 1—figure 1*, *Appendix 2—figure 1*), whereas *Tilman et al., 2020* reported the opposite conclusion. Furthermore, we observe that a small amplitude oscillation is more conducive to maintaining the stability of the system than a large magnitude oscillation because a higher risk will make it easier for all individuals to lose all their endowments.

The reciprocal feedback process, though many types have not been well characterized, occurs at all levels of our life (*Liu et al., 2007*; *Ezenwa et al., 2016*; *Obradovich and Rahwan, 2019*). Consequently, they may play an indispensable role in maintaining the stability of human society and the ecosystem. Mathematical modeling based on evolutionary game theory is a powerful tool for addressing social–ecological and human–environment interactions and analyzing the evolutionary dynamics of these coupled systems. The mathematical framework proposed in this article considers two characteristic forms to describe the effect of strategy on risk, namely, linear and nonlinear (exponential) forms of feedback. Although these two forms can be equivalent under some limit conditions, there are essential differences. On the one hand, linear relationship is a relatively simple way to describe the correlation mode of two factors, which is common in real society. For example, with the increase in protection awareness and vaccination proportion, the mortality rate of the epidemic decreased gradually (*Yang and Shaman, 2022*). Furthermore, linear feedback has been used to

describe the interactions between actions of the population and environmental state (*Weitz et al., 2016*; *Tilman et al., 2020*). However, linear link cannot fully describe the relationship between variables in real societies. For example, in recent years, extreme weather phenomena have occurred more frequently, with greater intensity and wider impact areas. Thus the feedback between human behaviors and environment may take on a more complex nonlinear form. In this work, we consider that the strategy of the population has an exponential effect on risk level, and such form can describe the phenomenon that risk will rise and fall sharply with the change in strategy frequency (*Figure 2*). It is worth emphasizing that although we use different forms of feedback to describe the impact of strategies on risk, the evolutionary dynamics have not changed substantially, which highlights the prime importance of the feedback mechanism independently of its actual form.

Our feedback-evolving game model reveals that the coupled strategy and environment system will produce a variety of representative dynamical behaviors. We find that the undesired equilibrium point $(0, 1)$ in our feedback system is always evolutionarily stable, which does not depend on whether the effect of strategy on risk is linear or exponential. Such evolutionary outcome means that all individuals are unwilling to contribute to achieving the collective goal, which leads to the failure of collective action, and all individuals inevitably lose their remaining endowments. In real-world scenarios, such as climate change (*Milinski et al., 2008*) and the spread of infectious diseases (*Cronk and Aktipis, 2021*; *Chen and Fu, 2022*), once the whole society is in such a state, it is undoubtedly disastrous for the public. Therefore, how to adjust and control the system to deviate from this state is particularly important for policymakers.

Finally, it is worth emphasizing that the feedback loop operates over time. In this situation, the change in risk state or strategy frequency may lead to the change in other factors, such as collective target, which provides an opportunity for the emergence of new feedback loops. Thus, multiple types of feedback loops are possible in a single coupled system. Such multiple feedback loops have been confirmed in the coupling system of animal behavior and disease ecology (*Ezenwa et al., 2016*). Therefore, a promising expansion of our current model could be to consider the multiple feedback loops.

## Code availability

The Mathematica (Wolfram Mathematica 11.1) source code used to generate *Figures 4 and 6* is available on GitHub (copy archived at *Liu, 2023*).

## Acknowledgements

This research was supported by the National Natural Science Foundation of China (grant nos. 61976048 and 62036002) and the Fundamental Research Funds of the Central Universities of China. LL acknowledges the support from Special Project of Scientific and Technological Innovation (grant no. 2452022012) and the Natural Science Foundation of Shaanxi Province (grant no. JC-QN-0791). AS was supported by the National Research, Development and Innovation Office (NKFIH), Hungary under grant no. K142948.

## Additional information

### Funding

| Funder | Grant reference number | Author |
|---|---|---|
| National Natural Science Foundation of China | 61976048 | Xiaojie Chen |
| National Natural Science Foundation of China | 62036002 | Xiaojie Chen |
| Natural Science Foundation of Shaanxi Province | JC-QN-0791 | Linjie Liu |

| Funder | Grant reference number | Author |
| --- | --- | --- |
| National Research, Development and Innovation Office | K142948 | Attila Szolnoki |

The funders had no role in study design, data collection and interpretation, or the decision to submit the work for publication.

## Author contributions

Linjie Liu, Conceptualization, Formal analysis, Validation, Investigation, Methodology, Writing – original draft, Writing – review and editing; Xiaojie Chen, Conceptualization, Formal analysis, Supervision, Writing – review and editing; Attila Szolnoki, Formal analysis, Writing – review and editing

## Author ORCIDs

Linjie Liu http://orcid.org/0000-0003-0286-8885
Xiaojie Chen http://orcid.org/0000-0002-9129-2197
Attila Szolnoki http://orcid.org/0000-0002-0907-0406

## Decision letter and Author response

Decision letter https://doi.org/10.7554/eLife.82954.sa1
Author response https://doi.org/10.7554/eLife.82954.sa2

# Additional files

## Supplementary files

• MDAR checklist

## Data availability

The Mathematica (Wolfram Mathematica 11.1) and Matlab (Matlab R2014a) source codes used to generate Figures 4 and 6 are available on Dryad.

The following dataset was generated:

| Author(s) | Year | Dataset title | Dataset URL | Database and Identifier |
| --- | --- | --- | --- | --- |
| Liu L, Chen X, Szolnoki A | 2023 | Code for: Coevolutionary dynamics via adaptive feedback in collective-risk social dilemma game | https://doi.org/10.5061/dryad.wdbrv15rz | Dryad Digital Repository, 10.5061/dryad.wdbrv15rz |

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

## Appendix 1

We first study the case where the strategy of the population has a linear effect on the risk level. Then the dynamical system can be written as

$$\begin{cases} \varepsilon\dot{x} = x(1-x)[\binom{N-1}{M-1}x^{M-1}(1-x)^{N-M}rb - c], \\[2mm] \dot{r} = r(1-r)[u(1-x) - x]. \end{cases}$$

This equation system has at most seven fixed points, which are $(0,0)$, $(0,1)$, $(1,0)$, $(1,1)$, $(\frac{u}{1+u}, \frac{c}{\binom{N-1}{M-1}(\frac{u}{1+u})^{M-1}(\frac{1}{1+u})^{N-M}b})$, $(x_1^*, 1)$ and $(x_2^*, 1)$, where $x_1^*$ and $x_2^*$ are the real roots of the equation $\binom{N-1}{M-1}x^{M-1}(1-x)^{N-M}b = c$. For convenience, we introduce the abbreviation $r^* = \frac{c}{\binom{N-1}{M-1}(\frac{u}{1+u})^{M-1}(\frac{1}{1+u})^{N-M}b}$ and $\Gamma(x) = \binom{N-1}{M-1}x^{M-1}(1-x)^{N-M}$. In the following, we analyze the stability of these equilibrium points.

(1) When $0 < r^* < 1$, namely, $\Gamma(\frac{u}{1+u}) > \frac{c}{b}$, the system has an interior fixed point. Accordingly, the Jacobian for the interior fixed point is

$$J(\frac{u}{1+u}, r^*) = \begin{bmatrix} \bar{a}_{11} & \bar{a}_{12} \\ \bar{a}_{21} & 0 \end{bmatrix},$$

where $\bar{a}_{11} = \frac{c}{\varepsilon}[M - 1 - \frac{u(N-1)}{u+1}]$, $\bar{a}_{12} = \frac{1}{\varepsilon}\binom{N-1}{M-1}(\frac{u}{1+u})^M(\frac{1}{1+u})^{N-M+1}b$, and $\bar{a}_{21} = -r^*(1-r^*)(1+u)$.

(i) When $\bar{a}_{11} > 0$, namely, $u < \frac{M-1}{N-M}$, the existing interior fixed point is unstable. Since $\Gamma(\frac{M-1}{N-1}) > \frac{c}{b}$, we can know that the two boundary fixed points $(x_1^*, 1)$ and $(x_2^*, 1)$ exist. Thus, the system has seven fixed points in the parameter space, namely, $(0,0)$, $(0,1)$, $(1,0)$, $(1,1)$, $(\frac{u}{1+u}, r^*)$, $(x_1^*, 1)$, and $(x_2^*, 1)$. The Jacobian matrices of these equilibrium points are respectively given as follows.

For $(x, r) = (0, 0)$, the Jacobian is

$$J(0,0) = \begin{bmatrix} -\frac{c}{\varepsilon} & 0 \\ 0 & u \end{bmatrix},$$

thus the fixed equilibrium is unstable.

For $(x, r) = (0, 1)$, the Jacobian is

$$J(0,1) = \begin{bmatrix} -\frac{c}{\varepsilon} & 0 \\ 0 & -u \end{bmatrix},$$

thus the fixed equilibrium is stable.

For $(x, r) = (1, 0)$, the Jacobian is

$$J(1,0) = \begin{bmatrix} \frac{c}{\varepsilon} & 0 \\ 0 & -1 \end{bmatrix},$$

thus the fixed equilibrium is unstable.

For $(x, r) = (1, 1)$, the Jacobian is

$$J(1,1) = \begin{bmatrix} \frac{c}{\varepsilon} & 0 \\ 0 & 1 \end{bmatrix},$$

thus the fixed equilibrium is unstable.

For $(x, r) = (x_1^*, 1)$, the Jacobian is

$$J(x_1^*, 1) = \begin{bmatrix} \frac{c}{\varepsilon}(M - 1 - x_1^*(N - 1)) & \frac{c}{\varepsilon}x_1^*(1 - x_1^*) \\ 0 & (1 + u)x_1^* - u \end{bmatrix},$$

thus the fixed equilibrium is unstable since $x_1^* < \frac{M-1}{N-1}$.

For $(x, r) = (x_2^*, 1)$, the Jacobian is

$$J(x_2^*, 1) = \begin{bmatrix} \frac{c}{\varepsilon}(M - 1 - x_2^*(N - 1)) & \frac{c}{\varepsilon}x_2^*(1 - x_2^*) \\ 0 & (1 + u)x_2^* - u \end{bmatrix},$$

because $u < \frac{M-1}{N-M}$ and $x_2^* > \frac{M-1}{N-1}$, then $1 - \frac{1}{1+u} < \frac{M-1}{N-1} < x_2^*$. Thus this fixed equilibrium is unstable.

(ii) When $\bar{a}_{11} = 0$, namely, $u = \frac{M-1}{N-M}$, the trace and determinant of the Jacobian matrix at the interior equilibrium point are respectively given by

$$\text{tr}(J(\tfrac{u}{1+u}, r^*)) = \bar{a}_{11} = 0,$$
$$\det(J(\tfrac{u}{1+u}, r^*)) = -\bar{a}_{12}\bar{a}_{21} = \frac{r^*(1-r^*)(1+u)}{\varepsilon}\binom{N-1}{M-1}(\tfrac{u}{1+u})^M(\tfrac{1}{1+u})^{N-M+1}b > 0.$$

The eigenvalues of the Jacobian matrix can be calculated

$$\lambda_1 = \frac{\bar{a}_{11} + \sqrt{\bar{a}_{11}^2 + 4\bar{a}_{12}\bar{a}_{21}}}{2} = \bar{\mu} + i\bar{w},$$
$$\lambda_2 = \frac{\bar{a}_{11} - \sqrt{\bar{a}_{11}^2 + 4\bar{a}_{12}\bar{a}_{21}}}{2} = \bar{\mu} - i\bar{w},$$

where $\bar{\mu} = \frac{\bar{a}_{11}}{2} = 0$ and $\bar{w}^2 = -\bar{a}_{12}\bar{a}_{21}$.

Accordingly, we know that the eigenvalues satisfy the following conditions:

$$\begin{aligned} \text{Re}(\lambda) &= \bar{\mu} = 0, \\ \text{Im}(\lambda) &= \frac{\sqrt{-\bar{a}_{11}^2 - 4\bar{a}_{12}\bar{a}_{21}}}{2} \neq 0, \\ \frac{d\text{Re}(\lambda)}{du}\big|_{u = \frac{M-1}{N-M}} &= -\frac{c(N-1)}{2\varepsilon(u+1)^2} = -\frac{c(N-M)^2}{2\varepsilon(N-1)} < 0. \end{aligned}$$

The first two conditions imply that the eigenvalues of Jacobian matrix at $(\frac{u}{1+u}, r^*)$ has a pair of pure imaginary roots. The third condition means that the pair of complex-conjugate eigenvalues crosses the imaginary axis with nonzero speed. According to Hopf bifurcation theorem (**Kuznetsov, 1998**), we know that a Hopf bifurcation takes place at $u = \frac{M-1}{N-M}$. In order to determine the stability of the existing limit cycle from Hopf bifurcation, we need to calculate the first Lyapunov coefficient. We denote that $F_1(x, r) = \frac{x(1-x)}{\varepsilon}[\binom{N-1}{M-1}x^{M-1}(1-x)^{N-M}rb - c]$ and $F_2(x, r) = r(1-r)[u(1-x) - x]$.

Let $q, p \in \mathbb{C}^2$ respectively denote the eigenvectors of the Jacobian matrix $J(T, r^*)$ and its transpose. We then have

$$q = \begin{pmatrix} \frac{-i\bar{a}_{12}}{\bar{w}} \\ 1 \end{pmatrix}, \quad p = \begin{pmatrix} \frac{-i\bar{w}}{\bar{a}_{12}} \\ 1 \end{pmatrix}, \tag{8}$$

which satisfy

$$\begin{aligned} Jq &= i\bar{w}q, \\ J^T p &= -i\bar{w}p. \end{aligned}$$

To achieve the necessary normalization $< p, q > = \bar{p}_1 q_1 + \bar{p}_2 q_2 = 1$, we can take

$$q = \begin{pmatrix} \frac{-i\bar{a}_{12}}{2\bar{w}} \\ \frac{1}{2} \end{pmatrix}, \quad p = \begin{pmatrix} \frac{-i\bar{w}}{\bar{a}_{12}} \\ 1 \end{pmatrix}, \tag{9}$$

According to **Kuznetsov, 1998**, we construct the complex-valued function

$$G(x, r) = \bar{p}_1 F_1(\tfrac{u}{1+u} + xq_1 + r\bar{q}_1, r^* + xq_2 + r\bar{q}_2) + \bar{p}_2 F_2(\tfrac{u}{1+u} + xq_1 + r\bar{q}_1, r^* + xq_2 + r\bar{q}_2),$$

where $p, q$ are given above, to evaluate its formal partial derivatives with respect to $x, r$ at $(T, r^*)$, obtaining $g_{20} = G_{xx}, g_{11} = G_{xr}$, and $g_{21} = G_{xxr}$. After some calculations, we can get the first Lyapunov coefficient

$$l_1 = \frac{1}{2\bar{w}^2} \mathrm{Re}(ig_{20}g_{11} + \bar{w}g_{21}).$$

Specifically, when $l_1 < 0$, a unique and stable limit cycle bifurcates from the equilibrium appears, while when $l_1 > 0$, the Hopf bifurcation is subcritical such that an unstable limit cycle will be generated. Due to the complexity of the system, it is difficult to conduct bifurcation analysis collectively. Here, we conduct a numerical analysis to investigate the stability of the existing limit cycle when the model parameters are consistent with **Figure 4d**. By using the algorithm in **Kuznetsov, 1998**, we can get $l_1 = -1.407166124 \times 10^{-8} < 0$, which implies that the Hopf bifurcation is supercritical.

Besides, since $\Gamma(\frac{M-1}{N-1}) > \frac{c}{b}$, we can state that the two boundary fixed points $(x_1^*, 1)$ and $(x_2^*, 1)$ exist. Thus the system has seven equilibrium points, which are $(0, 0)$, $(0, 1)$, $(1, 0)$, $(1, 1)$, $(\frac{u}{1+u}, r^*)$, $(x_1^*, 1)$, and $(x_2^*, 1)$, respectively. Accordingly to the sign of the eigenvalues of the Jacobian matrices, we know that only $(0, 1)$ is stable.

(iii) When $\bar{a}_{11} < 0$, namely, $u > \frac{M-1}{N-M}$, the trace and determinant of the Jacobian matrix at the interior equilibrium point are respectively given by

$$\mathrm{tr}(J(\tfrac{u}{1+u}, r^*)) = \bar{a}_{11} < 0,$$

$$\det(J(\tfrac{u}{1+u}, r^*)) = -\bar{a}_{12}\bar{a}_{21} = \frac{r^*(1-r^*)(1+u)}{\varepsilon} \binom{N-1}{M-1} (\frac{u}{1+u})^M (\frac{1}{1+u})^{N-M+1} b > 0.$$

Thus the interior fixed point is stable. Besides, since $\Gamma(\frac{M-1}{N-1}) > \frac{c}{b}$, two boundary fixed points, $(x_1^*, 1)$ and $(x_2^*, 1)$, exist. Thus there are seven fixed points in the system, which are $(0, 0)$, $(0, 1)$, $(1, 0)$, $(1, 1)$, $(\frac{u}{1+u}, r^*)$, $(x_1^*, 1)$, and $(x_2^*, 1)$, respectively. Here, the fixed points $(0, 1)$ and $(\frac{u}{1+u}, r^*)$ are stable, while others are unstable.

(2) When $r^* \geq 1$, namely, $\Gamma(\frac{u}{1+u}) \leq \frac{c}{b}$, the system has no interior equilibrium point. In this case, when $\Gamma(\frac{M-1}{N-1}) > \frac{c}{b}$, the system has six fixed points, which are $(0, 0)$, $(0, 1)$, $(1, 0)$, $(1, 1)$, $(x_1^*, 1)$, and $(x_2^*, 1)$, respectively. According to the sign of the largest eigenvalues of the Jacobian matrices, we can say that $(0, 0)$, $(1, 0)$, $(1, 1)$, $(x_1^*, 1)$ are unstable, while $(0, 1)$ is stable. Particularly, when $x_2^* < \frac{u}{1+u}$, the fixed point $(x_2^*, 1)$ is stable, and it is unstable when $x_2^* > \frac{u}{1+u}$. When $x_2^* = \frac{u}{1+u}$, we know that one eigenvalue of the Jacobian matrix is zero and the other eigenvalue is negative. Then we study its stability by using the center manifold theorem (**Khalil, 1996**). For the fixed point $(x_2^*, 1)$, the Jacobian matrix can be written as

$$J(x_2^*, 1) = \begin{bmatrix} \gamma_{11} & \gamma_{12} \\ 0 & 0 \end{bmatrix},$$

where $\gamma_{11} = \frac{c}{\varepsilon}(M - 1 - x_2^*(N - 1))$ and $\gamma_{12} = \frac{c}{\varepsilon}x_2^*(1 - x_2^*)$. To do that, we take $z_1 = x - x_2^*$ and $z_2 = r - 1$, then the system can be rewritten as

$$\begin{cases} \dot{z}_1 = \frac{1}{\varepsilon}(x_2^* + z_1)(1 - x_2^* - z_1)[\binom{N-1}{M-1}(x_2^* + z_1)^{M-1}(1 - x_2^* - z_1)^{N-M}(z_2 + 1)b - c], \\ \dot{z}_2 = (z_2 + 1)(-z_2)[u(1 - x_2^* - z_1) - x_2^* - z_1]. \end{cases}$$

Let $Q$ be a matrix whose columns are the eigenvectors of $J(x_2^*, 1)$, which can be written as

$$Q = \begin{bmatrix} 1 & -\frac{\gamma_{12}}{\gamma_{11}} \\ 0 & 1 \end{bmatrix}.$$

Then we have

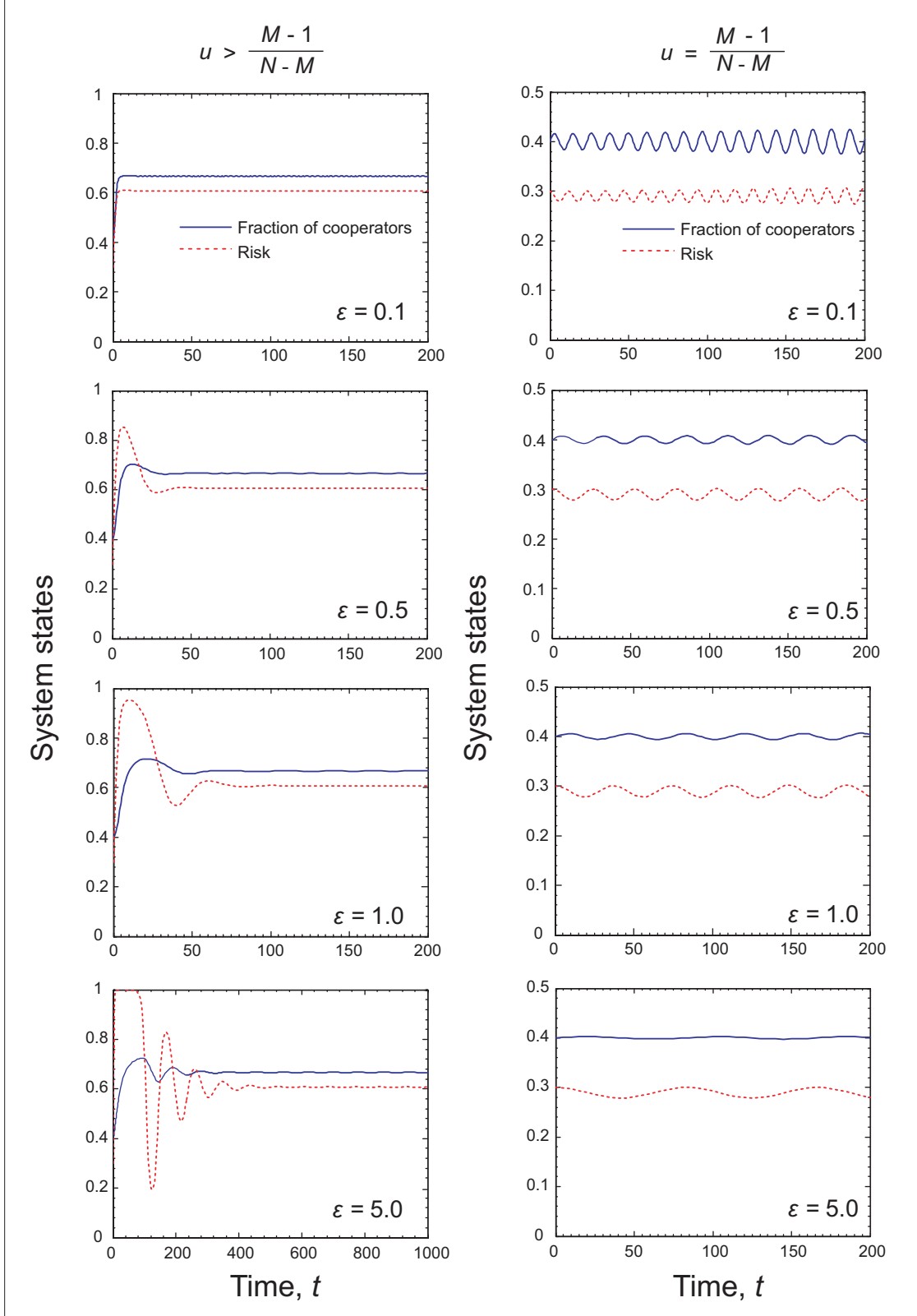

**Appendix 1—figure 1.** Coevolutionary dynamics of System I for different $\varepsilon$ values when linear feedback effect of strategy on risk level is considered. Parameters are $N = 6, c = 0.1, b = 1$, and $M = 3$ in left column and $u = 2/3$ in right column. The initial conditions are $(x, r) = (0.4, 0.3)$.

$$Q^{-1}JQ = \begin{bmatrix} \gamma_{11} & 0 \\ 0 & 0 \end{bmatrix}.$$

We further take $[\eta_1 \quad \eta_2]^T = Q^{-1}[z_1 \quad z_2]$, and then we have $\eta_1 = z_1 + \frac{\gamma_{12}}{\gamma_{11}}z_2$ and $\eta_2 = z_2$. We then have

$$\dot{\eta}_2 = -\eta_2(\eta_2+1)[u(1 - \frac{u}{1+u} - \eta_1 + \frac{\gamma_{12}}{\gamma_{11}}\eta_2) - \frac{u}{1+u} - \eta_1 + \frac{\gamma_{12}}{\gamma_{11}}\eta_2].$$

According to the center manifold theorem, we know that $\eta_1 = h(\eta_2)$ is a center manifold. Then we start to try $h(\eta_2) = O(|\eta_2|^2)$, which yields the reduced system

$$\dot{\eta}_2 = -(1+u)\frac{\gamma_{12}}{\gamma_{11}}\eta_2^2 - (1+u)\frac{\gamma_{12}}{\gamma_{11}}\eta_2^3 + O(|\eta_2|^4).$$

Since $-(1+u)\frac{\gamma_{12}}{\gamma_{11}} \neq 0$, the fixed point $\eta_2 = 0$ of the reduced system is unstable. Accordingly, the fixed point $(x_2^*, 1)$ of the original system is unstable.

When $\Gamma(\frac{M-1}{N-1}) = \frac{c}{b}$, the system has five fixed points, which are $(0,0), (0,1), (1,0), (1,1)$, and $(\frac{M-1}{N-1}, 1)$, respectively. According to the sign of the eigenvalues in the Jacobian matrices, we can state that only $(0,1)$ is stable. When $\Gamma(\frac{M-1}{N-1}) < \frac{c}{b}$, the system has four fixed points, namely, $(0,0), (0,1), (1,0)$, and $(1,1)$. Here, only $(0,1)$ is stable.

# Appendix 2

System II with exponential feedback is described by

$$
\begin{cases}
\varepsilon \dot{x} = x(1-x)[\binom{N-1}{M-1} x^{M-1}(1-x)^{N-M} rb - c], \\
\dot{r} = r(1-r)[\dfrac{1}{1+e^{\beta(x-T)}} - \dfrac{1}{1+e^{-\beta(x-T)}}].
\end{cases}
$$

where the parameter $\beta > 0$ represents the steepness of the function.

This equation system has at most seven fixed points, which are $(0,0)$, $(0,1)$, $(1,0)$, $(1,1)$, $(T, \frac{c}{\binom{N-1}{M-1} T^{M-1}(1-T)^{N-M}b})$, $(x_1^*, 1)$ and $(x_2^*, 1)$, where $x_1^*$ and $x_2^*$ are the real roots of the equation $\binom{N-1}{M-1} x^{M-1}(1-x)^{N-M} b = c$ and $x_1^* < \frac{M-1}{N-1} < x_2^*$. For simplicity, we introduce the abbreviation $\bar{r} = \frac{c}{\binom{N-1}{M-1} T^{M-1}(1-T)^{N-M}b}$ and $\Gamma(x) = \binom{N-1}{M-1} x^{M-1}(1-x)^{N-M}$. In the following, we study the stabilities of equilibria based on whether the system has an interior equilibrium point.

(1) When $0 < \bar{r} < 1$, namely, $\Gamma(T) > \frac{c}{b}$, System II has an interior equilibrium point.

The Jacobian matrix evaluated at this equilibrium is

$$
J(T, \bar{r}) =
\begin{bmatrix}
a_{11} & a_{12} \\
a_{21} & 0
\end{bmatrix},
$$

where $a_{11} = \frac{c}{\varepsilon}(M-1-T(N-1))$, $a_{12} = \frac{1}{\varepsilon}\binom{N-1}{M-1} T^{M}(1-T)^{N-M+1}b$, and $a_{21} = -\frac{\bar{r}(1-\bar{r})\beta}{2}$. Notice that $\frac{1}{\varepsilon}\binom{N-1}{M-1} T^{M}(1-T)^{N-M+1}b > 0$ and $-\frac{\bar{r}(1-\bar{r})\beta}{2} < 0$, then the trace and determinant of the Jacobian matrix are respectively given by

$$
\text{tr}(J(T, \bar{r})) = \frac{c}{\varepsilon}(M-1-T(N-1)),
$$
$$
\det(J(T, \bar{r})) = \frac{1}{\varepsilon}\binom{N-1}{M-1} T^{M}(1-T)^{N-M+1}b \frac{\bar{r}(1-\bar{r})\beta}{2} > 0.
$$

The eigenvalues of the Jacobian matrix can be calculated as

$$
\lambda_1 = \frac{a_{11} + \sqrt{a_{11}^2 + 4a_{12}a_{21}}}{2},
$$
$$
\lambda_2 = \frac{a_{11} - \sqrt{a_{11}^2 + 4a_{12}a_{21}}}{2}.
$$

Here, we set that $\mu(T) = \frac{a_{11}}{2}$, $w^2(T) = -\frac{a_{11}^2 + 4a_{12}a_{21}}{4}$, and $T_0 = \frac{M-1}{N-1}$.

(i) When $a_{11} > 0$, namely, $T < T_0$, the interior equilibrium point is unstable. Since $\Gamma(T_0) > \frac{c}{b}$, we can know that the two boundary fixed points $(x_1^*, 1)$ and $(x_2^*, 1)$ exist. Thus, the system has seven fixed points in the parameter space, namely, $(0,0), (0,1), (1,0), (1,1), (T, \bar{r}), (x_1^*, 1)$, and $(x_2^*, 1)$.

For $(x, r) = (0, 0)$, the Jacobian is

$$
J(0,0) =
\begin{bmatrix}
-\frac{c}{\varepsilon} & 0 \\
0 & \frac{1-e^{-\beta T}}{1+e^{-\beta T}}
\end{bmatrix},
$$

thus the fixed equilibrium is unstable.

For $(x, r) = (0, 1)$, the Jacobian is

$$
J(0,1) =
\begin{bmatrix}
-\frac{c}{\varepsilon} & 0 \\
0 & -\frac{1-e^{-\beta T}}{1+e^{-\beta T}}
\end{bmatrix},
$$

thus the equilibrium point is stable.

For $(x, r) = (1, 0)$, the Jacobian is

$$J(1,0) = \begin{bmatrix} \frac{c}{\varepsilon} & 0 \\ 0 & \frac{1-e^{\beta(1-T)}}{1+e^{\beta(1-T)}} \end{bmatrix},$$

thus the fixed point is unstable.

For $(x, r) = (1, 1)$, the Jacobian is

$$J(1,1) = \begin{bmatrix} \frac{c}{\varepsilon} & 0 \\ 0 & -\frac{1-e^{\beta(1-T)}}{1+e^{\beta(1-T)}} \end{bmatrix},$$

thus the fixed equilibrium is unstable.

For $(x, r) = (x_1^*, 1)$, the Jacobian is

$$J(x_1^*, 1) = \begin{bmatrix} \frac{c}{\varepsilon}(M - 1 - x_1^*(N-1)) & \frac{c}{\varepsilon}x_1^*(1 - x_1^*) \\ 0 & -\frac{1-e^{\beta(x_1^* - T)}}{1+e^{\beta(x_1^* - T)}} \end{bmatrix},$$

thus the fixed equilibrium is unstable since $x_1^* < T_0$.

For $(x, r) = (x_2^*, 1)$, the Jacobian is

$$J(x_2^*, 1) = \begin{bmatrix} \frac{c}{\varepsilon}(M - 1 - x_2^*(N-1)) & \frac{c}{\varepsilon}x_2^*(1 - x_2^*) \\ 0 & -\frac{1-e^{\beta(x_2^* - T)}}{1+e^{\beta(x_2^* - T)}} \end{bmatrix},$$

thus the fixed equilibrium is unstable since $T < T_0 < x_2^*$.

(ii) When $a_{11} = 0$, namely, $T = T_0 = \frac{M-1}{N-1}$, we have $\mu(T_0) = 0$. Moreover, $w^2(T) = -\frac{a_{11}^2 + 4a_{12}a_{21}}{4} = \frac{1}{\varepsilon}\binom{N-1}{M-1}T^M(1-T)^{N-M+1}b\frac{\bar{r}(1-\bar{r})\beta}{2} > 0$. Therefore, the eigenvalues of the Jacobian matrix are a purely imaginary conjugate pair $\lambda_{1,2}(T_0) = \pm iw(T_0)$. Considering that $\frac{\partial\mu(T)}{\partial T}|_{T_0} = -\frac{c(N-1)}{2\varepsilon} < 0$, then we know that the system undergoes a Hopf bifurcation at $T = T_0$ and there exists a limit cycle around the interior equilibrium. Accordingly, we can evaluate the direction of the limit cycle bifurcation by computing the first Lyapunov coefficient $l_1$ of the system. Here, we also conduct numerical calculations to investigate the stability of the existing limit cycle when the model parameters are consistent with *Figure 6d*. By using the algorithm in *Kuznetsov, 1998*, we can get $l_1 = -1.876221498 \times 10^{-8}$, which implies that the Hopf bifurcation is supercritical.

Besides, since $\Gamma(T_0) > \frac{c}{b}$, we know that there are seven equilibrium points in System II. They are $(0,0)$, $(0,1)$, $(1,0)$, $(1,1)$, $(T, \bar{r})$, $(x_1^*, 1)$, and $(x_2^*, 1)$. According to the sign of the eigenvalues of the Jacobian matrices, only $(0,1)$ is stable.

(iii) When $a_{11} < 0$, namely, $T > T_0$, the interior equilibrium point is stable. Besides, since $\Gamma(T_0) > \frac{c}{b}$, we find that there are seven fixed points in the system, which are $(0,0)$, $(0,1)$, $(1,0)$, $(1,1)$, $(T, \bar{r})$, $(x_1^*, 1)$, and $(x_2^*, 1)$, respectively. Here, the fixed points $(0,1)$ and $(T, \bar{r})$ are stable, while others are unstable.

(2) When $\bar{r} \geq 1$, namely, $\Gamma(T) \leq \frac{c}{b}$, System II has no interior equilibrium point. In this case, when $\Gamma(T_0) > \frac{c}{b}$, the system has six fixed points, which are $(0,0), (0,1), (1,0), (1,1), (x_1^*, 1)$, and $(x_2^*, 1)$, respectively. According to the sign of the largest eigenvalues of the Jacobian matrices, we can say that $(0,0), (1,0), (1,1), (x_1^*, 1)$ are unstable, while $(0,1)$ is stable. Particularly, when $x_2^* < T$, the fixed point $(x_2^*, 1)$ is stable, and it is unstable when $x_2^* > T$. When $\Gamma(T_0) = \frac{c}{b}$, the system has five fixed points, which are $(0,0), (0,1), (1,0), (1,1)$, and $(T_0, 1)$, respectively. According to the sign of the eigenvalues in the Jacobian matrices, we can see that only $(0,1)$ is stable. When $\Gamma(T_0) < \frac{c}{b}$, the system has four fixed points, namely $(0,0), (0,1), (1,0)$, and $(1,1)$. Here, only $(0,1)$ is stable.

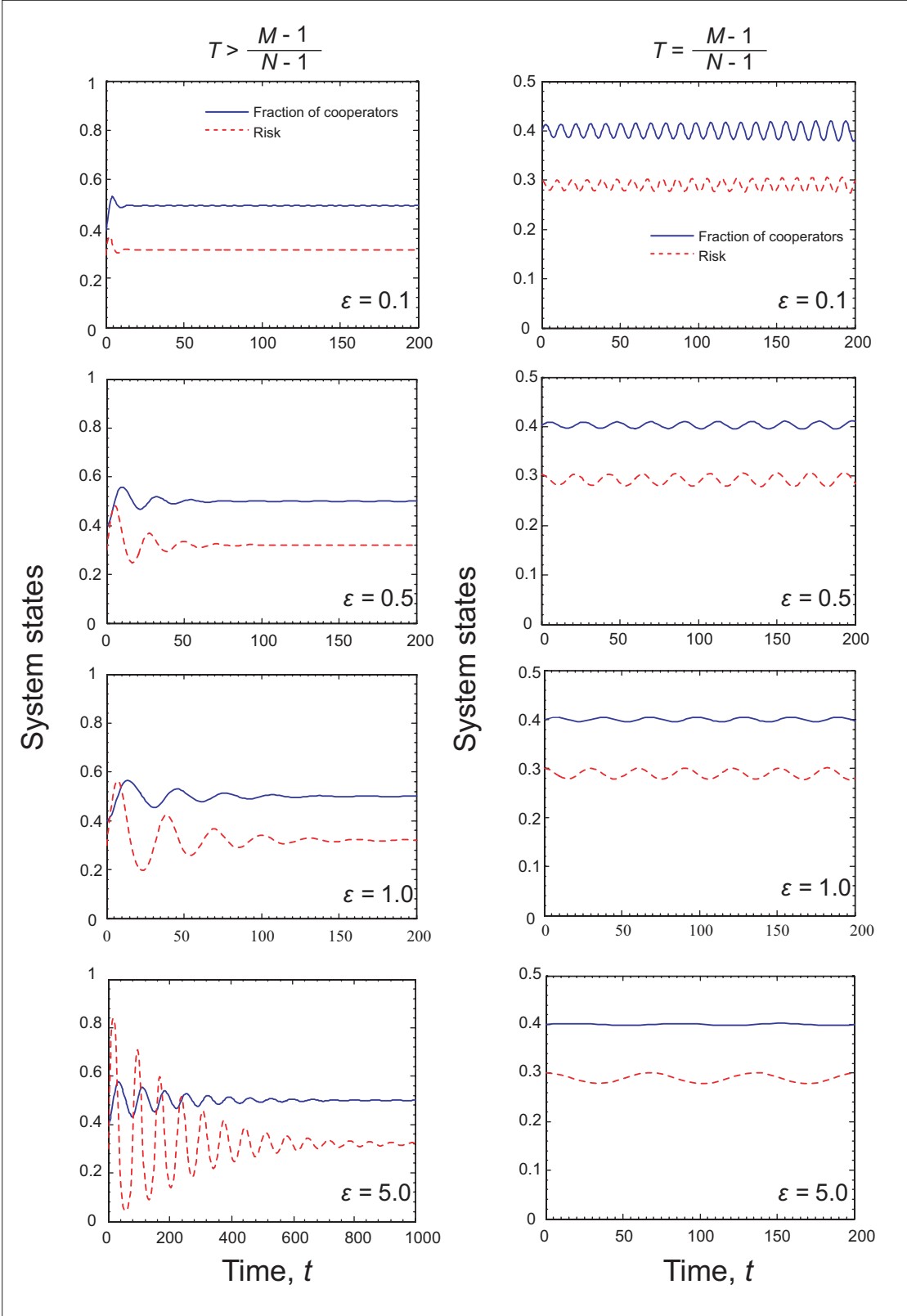

**Appendix 2—figure 1.** Coevolutionary dynamics of System II for different $\varepsilon$ values when the strategy feedback on risk is exponential. Parameters are $N = 6, c = 0.1, b = 1$, and $M = 3$ in the left column and $T = 0.4$ in the right column. The initial condition is $(x, r) = (0.4, 0.3)$.

