## [Editor Report]

The paper provides a valuable, in-depth mathematical analysis of the coevolutionary dynamics resulting from a coupling of players' strategies and (collective) risk, as well as illustrative numerical simulations of the system's trajectories for different starting conditions. It is therefore a solid contribution to our understanding of how cooperation can be sustained when there is feedback between individual decisions and the global risk of disaster. This paper will be of interest to scientists working on mathematical biology/ecology, and more generally various aspects of human decision-making, the interplay between human decisions and the environment, and public goods provision.

---

## [Decision Letter]

**Decision letter after peer review:**

Thank you for submitting your article "Coevolutionary dynamics via adaptive feedback in collective-risk social dilemma game" for consideration by *eLife*. Your article has been reviewed by 2 peer reviewers, and the evaluation has been overseen by a Reviewing Editor and Christian Rutz as the Senior Editor. The reviewers have opted to remain anonymous.

The reviewers have discussed their reviews with one another, and the Reviewing Editor has drafted this decision letter to help you prepare a revised submission.

Essential revisions:

1) A technical suggestion to improve the universality of their model.

As mentioned in the review, the authors regarded c/b as a parameter representing 'dilemma strength', which has been commonly adopted by many previous works. This is fine. Although we do not know if the authors have recognized this or not, there is an established idea; 'the concept of universal dilemma strength parameter for 2 by 2 games'. If they leave a PGG and come to a 2-by-2 game version, it is so-called Donor & Recipient (D & R) game, where T := b, R := b – c, P := -, and S := – c are imposed. Based on the concept, we quantify the Chicken-type dilemma; Dr' := (T – R) / (R – P) = c/(b – c) and Stag Hunt-type dilemma; Dr' := (P – S) / (R – P) = c/(b – c). Thus, a D & R game; 2 by 2 version of PGG, happens to have the same quantities of Dg' and Dr', which leads to a specific situation that the extent of social dilemma can be evaluated by a single model parameter; c / (b – c). Although we would not go so far as to say that they should replace 'c/b' by 'c/(b – c)', they should supplement in either Introduction or Model depiction part with the assertion that a PGG is a special and extended version of D & R game by referring to the concept of universal dilemma strength with citations to several relevant literatures; (i) Sociophysics Approach to Epidemics, Springer, 2021, (ii) Universal scaling for the dilemma strength in evolutionary games, Physics of Life Reviews 14, 1-30, 2015, (iii) Scaling the phase- planes of social dilemma strengths shows game-class changes in the five rules governing the evolution of cooperation, Royal Society Open Science, 181085, 2018.

2) While this is a solid paper, the presentation and flow could be improved. For one, it would be helpful to have a table with an overview of all the parameters and variables that are introduced over the course of the paper. Also, in the Results section, we believe it would help the flow and reading experience if there was more intuition about the actual interpretation of the equilibrium points. This means mentioning already earlier that (0,1) is the undesirable full defection equilibrium instead of only noting this in line 202, and explaining the meaning behind the other fixed points as well.

3) In terms of flow, it might also be useful to tighten up the Results and Discussion section considerably. In the former, some more technical details can be left out, given the very comprehensive appendix; furthermore, there are some redundant sentences (e.g., it would be enough to mention once that feedback speed does not influence the dynamics in any of the cases). This holds even more strongly for the Discussion: it is quite long-winded and would benefit from tightening and shortening, including a more concise summary of the results and less similarity with the Introduction, as well as removing redundant information where similar sentences appear in multiple paragraphs (e.g. about the robustness of the system to different feedback forms).

4) Finally, we think it would be interesting to compare the results of this paper in more depth with papers like (Hilbe, C., Šimsa, Š., Chatterjee, K., and Nowak, M. A. (2018). Evolution of cooperation in stochastic games. Nature, 446 559(7713):246-249), which was already cited in the manuscript but not expanded upon. The authors of that work analyze a repeated stochastic game with discrete time steps, which is quite different from the model considered in the present manuscript but has a similar aim of describing how cooperation can evolve when there is a two-way interplay between environment and strategy. Such a comparison also makes sense with regard to the sentence that " a two-way coupling…is essential to avoid the tragedy of the commons" (Abstract, line 22).

---

## [Author Response]

Essential revisions:1) A technical suggestion to improve the universality of their model.As mentioned in the review, the authors regarded c/b as a parameter representing 'dilemma strength', which has been commonly adopted by many previous works. This is fine. Although we do not know if the authors have recognized this or not, there is an established idea; 'the concept of universal dilemma strength parameter for 2 by 2 games'. If they leave a PGG and come to a 2-by-2 game version, it is so-called Donor & Recipient (D & R) game, where T := b, R := b – c, P := -, and S := – c are imposed. Based on the concept, we quantify the Chicken-type dilemma; Dr' := (T – R) / (R – P) = c/(b – c) and Stag Hunt-type dilemma; Dr' := (P – S) / (R – P) = c/(b – c). Thus, a D & R game; 2 by 2 version of PGG, happens to have the same quantities of Dg' and Dr', which leads to a specific situation that the extent of social dilemma can be evaluated by a single model parameter; c / (b – c). Although we would not go so far as to say that they should replace 'c/b' by 'c/(b – c)', they should supplement in either Introduction or Model depiction part with the assertion that a PGG is a special and extended version of D & R game by referring to the concept of universal dilemma strength with citations to several relevant literatures; (i) Sociophysics Approach to Epidemics, Springer, 2021, (ii) Universal scaling for the dilemma strength in evolutionary games, Physics of Life Reviews 14, 1-30, 2015, (iii) Scaling the phase- planes of social dilemma strengths shows game-class changes in the five rules governing the evolution of cooperation, Royal Society Open Science, 181085, 2018.

Thanks for the constructive suggestion. We have read them with great interest and found that all these mentioned works are conducive to reveal the role of dilemma strength in cooperation, and accordingly cited these relevant literatures in the Model section of the revised version.

2) While this is a solid paper, the presentation and flow could be improved. For one, it would be helpful to have a table with an overview of all the parameters and variables that are introduced over the course of the paper. Also, in the Results section, we believe it would help the flow and reading experience if there was more intuition about the actual interpretation of the equilibrium points. This means mentioning already earlier that (0,1) is the undesirable full defection equilibrium instead of only noting this in line 202, and explaining the meaning behind the other fixed points as well.

We thank the Reviewer for the valuable suggestions. We agree upon the Reviewer's point and we have added a table with an overview of all the parameters and variables that are introduced over the course of the paper in the revised version. In addition, we have added the actual interpretation of the equilibrium points in Result section to help the flow and reading experience.

3) In terms of flow, it might also be useful to tighten up the Results and Discussion section considerably. In the former, some more technical details can be left out, given the very comprehensive appendix; furthermore, there are some redundant sentences (e.g., it would be enough to mention once that feedback speed does not influence the dynamics in any of the cases). This holds even more strongly for the Discussion: it is quite long-winded and would benefit from tightening and shortening, including a more concise summary of the results and less similarity with the Introduction, as well as removing redundant information where similar sentences appear in multiple paragraphs (e.g. about the robustness of the system to different feedback forms).

We thank the Reviewer for his/her comments. We have removed some technical details and redundant sentences to tighten up the Results and Discussion section considerably. Concretely, we have removed the robustness of the system to different feedback forms and reduced the description of the impact of feedback speed on the evolutionary dynamics.

4) Finally, we think it would be interesting to compare the results of this paper in more depth with papers like (Hilbe, C., Šimsa, Š., Chatterjee, K., and Nowak, M. A. (2018). Evolution of cooperation in stochastic games. Nature, 446 559(7713):246-249), which was already cited in the manuscript but not expanded upon. The authors of that work analyze a repeated stochastic game with discrete time steps, which is quite different from the model considered in the present manuscript but has a similar aim of describing how cooperation can evolve when there is a two-way interplay between environment and strategy. Such a comparison also makes sense with regard to the sentence that " a two-way coupling…is essential to avoid the tragedy of the commons" (Abstract, line 22).

This is a great comment. We have added a paragraph in the Discussion section to compare the results of this paper in more depth with papers like (Hilbe, C., Šimsa, Š., Chatterjee, K., and Nowak, M. A. (2018). Evolution of cooperation in stochastic games. Nature, 446 559(7713):246-249). Indeed, we construct a quite different model, but have a similar aim of describing how cooperation can evolve when there is a two-way interplay between environment and strategy.